# A globally distributed dataset of coseismic landslide mapping via multi-source high-resolution remote sensing images

Chengyong Fang[1], Xuanmei Fan[1]*, Xin Wang[1], Lorenzo Nava[2], Hao Zhong[1,3], Xiujun Dong[1], Jixiao Qi[1], Filippo Catani[2]

[1]State Key Laboratory of Geohazard Prevention and Geoenvironment Protection, Chengdu University of Technology, 610059, Chengdu, China
[2]Machine Intelligence and Slope Stability Laboratory, Department of Geosciences, University of Padua, 35129 Padua, Italy
[3]College of Information Science and Technology, Chengdu University of Technology, 610059, Chengdu, China

*Correspondence to*: Xuanmei Fan (fxm_cdut@qq.com)

# Abstract

Rapid and accurate mapping of landslides triggered by extreme events is essential for effective emergency response, hazard mitigation, and disaster management. However, the development of generalized machine learning models for landslide detection has been hindered by the absence of a high-resolution, globally distributed, event-based dataset. To address this gap, we introduce the Globally Distributed Coseismic Landslide Dataset (GDCLD), a comprehensive dataset that integrates multi-source remote sensing images, including PlanetScope, Gaofen-6, Map World, and Unmanned Aerial Vehicle data, with varying geographical and geological background for nine events across the globe. In this study, we evaluated the effectiveness of GDCLD by comparing the mapping performance of seven state-of-the-art semantic segmentation algorithms. These models were further tested by three different types of remote sensing images in four independent regions, while the GDCLD-SegFormer model get the best performance. Additionally, we extended the evaluation to a rainfall-induced landslide dataset, where the models demonstrated excellent performance as well, highlighting the dataset's applicability to landslide segmentation triggered by other factors. Our results confirm the superiority of GDCLD in remote sensing landslide detection modeling, offering a comprehensive data base for rapid landslide assessment following future unexpected events worldwide.

# 1. Introduction

Landslides triggered by extreme events such as earthquakes and heavy precipitation are responsible for most of the damage to mountainous settlements (Huang and Fan, 2013). In some cases, landslides can be even more disastrous than the triggering events themselves, as they can render emergency responses ineffective by cutting off roads and other transportation lifelines (Cigna et al., 2012; Huang et al., 2012; Valagussa et al., 2019; Chau et al., 2004). Therefore, the rapid and accurate identification of landslides after extreme events is crucial for timely and quantitative assessment of disasters. This is especially important for emergency rescue operations and subsequent risk management in mountainous areas with complex environments and possibly inconvenient transportation routes. (Cigna et al., 2018; Chau et al., 2004; Gorum et al., 2011).

Conventional landslide mapping efforts rely on traditional surveying methods such as topographic total stations, field observations to collect essential data on slope stability and terrain morphology (Brardinoni et al., 2003; Coe et al., 2003; Zhong et al., 2020). These methods may not capture the full extent of terrain dynamics due to their static nature (Metternicht et al., 2005). Consequently, these methods are not effective for detailed landslide mapping, especially when traversing the affected and unstable regions for field surveys is not possible. This was particularly true for the Wenchuan co-seismic landslides, which mobilized large amounts of material that obstructed roads, complicating disaster response efforts as well as surveying and mapping activities (Gorum et al., 2011). With the development of remote sensing technology in the past decades, landslide investigation has been supported by digital mapping, which reduces time and labor costs (Fiorucci et al., 2011; Fiorucci et al., 2019; Gao and Maro, 2010; Guzzetti et al., 2012). This mapping has also been enhanced by various modalities of sensors, such as synthetic aperture radar (Mondini et al., 2021; Nava et al., 2021), multi-spectral (Udin et al., 2019), and hyper-spectral (Ye et al., 2019). However, visual identification is highly subjective due to operator experience, and the interpretation of events involving numerous landslides is still time-consuming. Therefore, this subjectivity and the time-

consuming nature of interpretation hinder the reliability and efficiency of landslide mapping, for
example, after major events such as the Wenchuan, China (2008), and Gorkha, Nepal (2015)
earthquakes.
Generally, the ideal solution is to develop automated models or tools that can save time
and costs while ensuring an objective protocol in the mapping process (Casagli et al., 2023).
While some researchers have endeavored to employ machine learning or deep learning in
constructing these models, most of them lack the generalization capability for application across
diverse environmental backgrounds and remote sensing images (Burrows et al., 2019; Bhuyan
et al., 2023; Li et al., 2016; Liu et al., 2022; Lu et al., 2019; Luppino et al., 2022; Meena et al.,
2021; Soares et al., 2022; Yang et al., 2022a; Mohan et al., 2021; Ss and Shaji, 2022; Li et al.,
2024). To improve such models, more abundant data that consider the diverse
geomorphological and climatic settings where landslides occur are essential. The Bijie landslide
dataset, based on Map World image, presents a small-scale dataset of mountainous landslides,
filling the gap in landslide detection tasks for the first time (Ji et al., 2020). Landslide4sense,
based on Sentinel-2 image, introduces a multispectral landslide dataset, pioneering semantic-
level annotation of landslides (Ghorbanzadeh et al., 2022). The HRGLDD and GVLM datasets,
based on PlanetScope and Google Earth image respectively, propose global-scale high-
resolution landslide datasets (Meena et al., 2022; Zhang et al., 2023). However, these datasets
are limited by their reliance on single remote sensing data sources, restricting the applicability
of models across different sensors and resolutions. The CAS dataset introduces a mountain
landslide dataset containing various remote sensing data sources (Xu et al., 2024). However,
due to its limited annotated landslide quantity, high image overlap, and lack of negative samples
(background/non-landslide), it is still insufficient to effectively generalize to landslide automatic
mapping tasks in various complex environments especially where signatures of landslides often
resemble nearby terrain.
Therefore, there is an urgent need to develop a carefully curated and diverse dataset. Such
a dataset would facilitate the rapid and accurate mapping of landslides using available prior
knowledge. Hence, we present a comprehensive landslide dataset derived from nine
earthquake-triggered landslide events, encompassing multi-sensor images from 3m-
PlanetScope, 2m-Gaofen-6, 0.5m-Map World, and 0.2m-UAV. This work addresses the
shortcomings of existing datasets in terms of accuracy and generalization for training large and
complex deep-learning models. It is of great significance for accurate, rapid, and automatic
mapping of landslides occurring anywhere in the world, providing strong support for efficient
geohazard emergency response and investigation.
The paper is structured as follows: Section 2 reviews existing high-quality landslide
datasets to provide an overview of the current state of research. Section 3 introduces the data
collection and preparation process to showcase the extensive research events and scientific
methodology behind our data production. Section 4 describes the semantic segmentation
algorithms, loss functions, and parameter settings used in this study, and shows their rationality.
Section 5 presents the results, including the training, validation, and testing outcomes of the
dataset, as well as the generalization ability of the GDCLD trained model in independent
regions. Section 6 discusses the innovation and effectiveness of GDCLD, illustrating its
effective application in several landslide events.

## 105  2. Related work

The most effective approach for landslide mapping currently involves image segmentation,
and computer vision segmentation tasks depend heavily on high-quality data to build accurate
models. However, landslide segmentation tasks have developed relatively recently compared
to other computer vision applications, resulting in only a limited number of studies that have
constructed datasets for various landslide events. In this section, we review some of these
landslide segmentation datasets and provide detailed information on each (Table.1).
The Bijie landslide dataset comprises high-resolution satellite images captured in
landslide-prone areas of Guizhou province, China. The dataset includes 770 landslide samples
and 2,003 non-landslide samples. The positive samples consist of rockfalls, rockslides, and a
small number of debris avalanches, while the negative samples include mountains, villages,
roads, rivers, and farmland, among others. The image resolutions vary from 61×61 pixels to
1,239×1,197 pixels, with RGB channels. There is a total of 7.23×10$^6$ pixels assigned for
landslide within the dataset (Ji et al., 2020).

The landslide4sense dataset consists of multispectral satellite images captured across four

distinct regions. This dataset comprises 3,799 images, each with dimensions of 64×64 pixels
and a spatial resolution of 10 meters. Each image contains 14 bands, including 12 bands from
the Sentinel-2 satellite and 2 bands from Digital Elevation Model (DEM) data. The dataset
includes negative background samples such as bare soil, rivers, and buildings. There is a total
of 1.76×10$^6$ pixels assigned for landslide within the dataset (Ghorbanzadeh et al., 2022).

The HR-GLDD spans 10 distinct geographic regions, capturing landslide instances across

various geographical environments in South Asia, Southeast Asia, East Asia, South America,
and Central America. HR-GLDD comprises a total of 1,756 image patches, each standardized
to a size of 128×128 pixels with a spatial resolution of up to 3 meters. The dataset is sourced
from four spectral bands of the PlanetScope satellite. It includes a variety of negative samples,
such as non-landslide terrain features, buildings, and roads, ensuring a comprehensive
representation for model training. There is a total of 2.96×10$^6$ pixels assigned for landslide
within the dataset (Meena et al., 2022).

The GVLM dataset spans across six continents and 17 different landslide sites, GVLM

covers a diverse range of geological and climatic conditions, from the lush landscapes of Asia
to the rugged terrain of South America. Comprising 17 pairs of dual-temporal VHR images,
each image pair boasts a spatial resolution of 0.59 meters, ensuring detailed capture of
landslide features and their surrounding environments. GVLM incorporates various negative
samples, including non-landslide landforms, infrastructure such as buildings, and transportation
networks, providing a holistic training ground for models. Image sizes within the GVLM dataset
range from 1,861×1,749 pixels to 10,828×7,424 pixels. There is a total of 3.24×10$^7$ pixels
assigned for landslide within the dataset (Zhang et al., 2023).

The CAS Landslide Dataset covers nine different geographic regions spanning South Asia,

Southeast Asia, East Asia, South America, and Central America. Comprising 20,865 image
patches, each standardized to a size of 512×512 pixels, the dataset offers a spatial resolution
ranging from 0.2 to 5 meters. During the cropping process, an overlap setting parameter of 0.5
was used. These images are sourced from unmanned aerial vehicles (UAVs) and satellite
platforms, integrating data from the PlanetScope satellite and other sources. The dataset
removes background images that do not contain landslide pixels and therefore lacks sufficient
background noise as negative samples to enhance the robustness of the model. There is a total
of $1.95 \times 10^8$ pixels assigned for landslide within the dataset (Xu et al., 2024).
In summary, comparing with other remote sensing detection tasks such as land cover/use,
the currently available landslide datasets are exceedingly scarce, predominantly comprising
single remote sensing images with low spatial resolutions. Overall, the available landslide
datasets are exceedingly limited, primarily comprising single remote sensing images with low
spatial resolution. Most crucially, these datasets lack sufficient annotations of landslide
instances, exhibit high overlap, and suffer from a dearth of diverse negative samples. As a
result, they are ill-equipped to tackle the challenges of mapping landslides in large-scale areas
with complex background objects, especially those sharing spectral and textural characteristics
with landslide surfaces, such as bare soil and rocks. Furthermore, they fail to provide adequate
data sources for effectively training large-scale neural network baseline models.
**Table.1** Existing landslide dataset statistics

| Dataset | Bands | events | Tiles | Landslides number | Labeling pixels |
|---|---|---|---|---|---|
| Bijie landslide | 3 | 1 | 2773 | 770 | $7.23 \times 10^6$ |
| Landslide4sense | 14 | 4 | 3799 | >30000 | $1.76 \times 10^6$ |
| HR-GLDD | 4 | 13 | 1756 | 7193 | $2.96 \times 10^6$ |
| GVLM | 3 | 17 | 17 | - | $3.24 \times 10^7$ |
| CAS Landslide | 3 | 9 | 20865 | - | $1.95 \times 10^8$ |

# 3. Globally Distributed Coseismic Landslide Dataset (GDCLD)

The creation of the GDCLD dataset can be broadly divided into two main components: landslide data collection and remote sensing data processing. In the first part, we compiled recent landslide events triggered by earthquakes worldwide over the past seven years and obtained the corresponding remote sensing image. The second part details the process of annotating landslide labels and the methodology used to create the standard dataset. The workflow is illustrated in Figure.1.

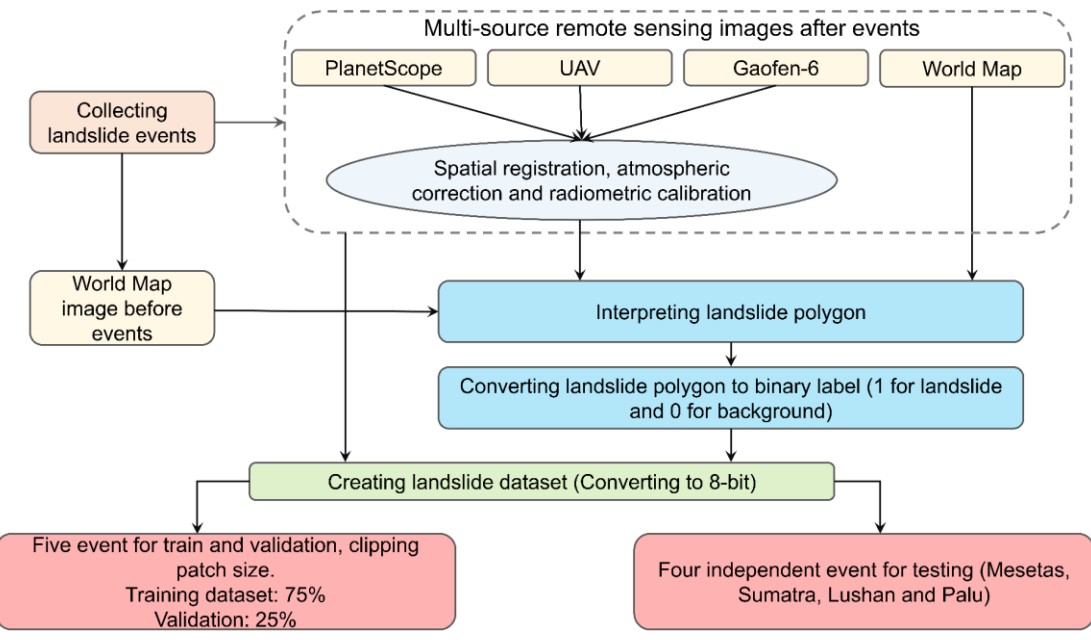

**Figure.1** The workflow of producing GDCLD

## 3.1 Data collection

Our dataset encompasses a catalog of landslides triggered by nine seismic occurrences, delineated across the Himalayan seismic belt and the Circum-Pacific belt, as depicted in Figure.2. These regions have witnessed actively seismic events with magnitudes over 5.9, triggering numerous landslides (Table.2). We obtained data of these locations from various remote sensing sources. This section delineates the particulars of the seismic events and the provenance of the remote sensing data.

Table.2 Summary table of landslide event information in GDCLD

| Events | Mw | time | Geographic coordinates | Landslide number | Total landslide area (km$^2$) |
|---|---|---|---|---|---|
| Jiuzhaigou | 6.5 | 2017 | (102.82°E, 33.20°N) | 2498 | 14.5 |
| Mainling | 6.4 | 2017 | (95.02°E, 29.75°N) | 1448 | 33.6 |
| Hokkaido | 6.6 | 2018 | (142.01°E, 42.69°N) | 7962 | 23.8 |
| Palu | 7.5 | 2018 | (119.84°E, 0.18°S) | 15700 | 43.0 |
| Mesetas | 6.0 | 2019 | (76.19°W, 3.45°N) | 804 | 8.5 |
| Nippes | 7.2 | 2021 | (73.45°W, 18.35°N) | 4893 | 45.6 |
| Sumatra | 6.1 | 2022 | (100.10°E, 0.22°N) | 602 | 10.6 |
| Lushan | 5.9 | 2022 | (102.94°E, 30.37°N) | 1063 | 7.2 |
| Luding | 6.8 | 2022 | (102.08°E, 29.59°N) | 15163 | 28.53 |

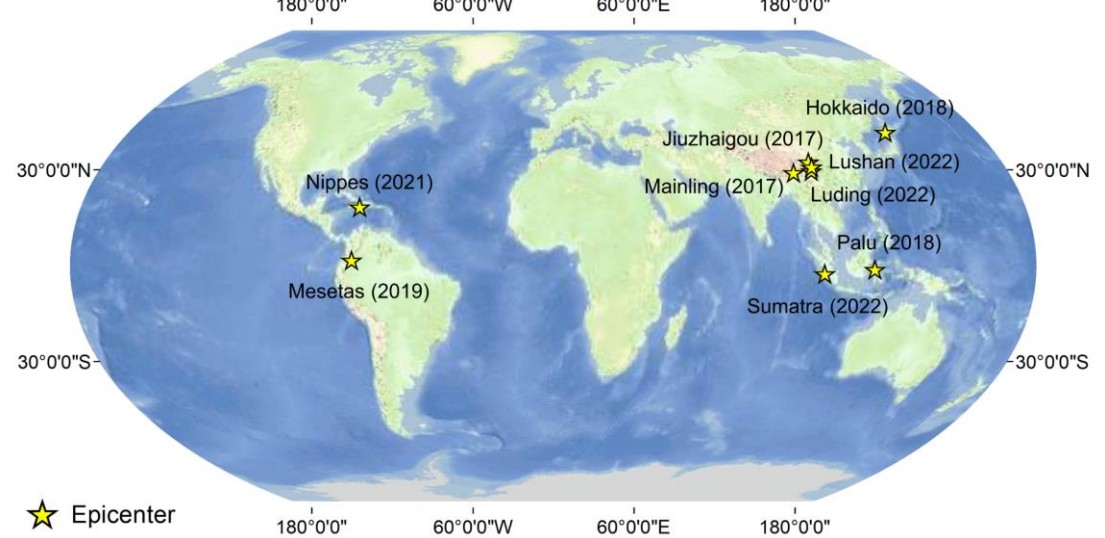


**Figure.2** Distribution of earthquake-triggerred landslide events

**3.1.1 The 2017 Jiuzhaigou earthquake-triggered landslides**

On August 8, 2017, a Mw 6.5 earthquake struck Jiuzhaigou County in Sichuan Province,

China (102.82°E, 33.20°N), triggering 2,498 landslides, predominantly shallow surface slides
and collapses. The largest landslide covered approximately 2.3×10$^5$m² (Fan et al., 2018).
Jiuzhaigou, situated on the northeastern margin of the Qinghai-Tibet Plateau within the
tectonically active zone north of the Longmenshan fault, is part of the Mediterranean Himalayan
seismic belt (Fan et al., 2018). The region's average elevation exceeds 3,000m with a maximum
relief of 2,228m and a vegetation cover surpassing 70% (Yi et al., 2020; Chen et al., 2019).
Exposed geological formations include various gray-white sandstones and dolomites from the
Devonian, Carboniferous, Permian, Triassic, and Tertiary periods (Fang et al., 2022). Post-
earthquake, we acquired multiple remote sensing images: 0.2m-resolution UAV image (Phase
One IXU1000) on September 22, 2017, 3m-resolution PlanetScope image on October 13, 2017,
and 0.5m-resolution from Map World (Figure.S1).
**3.1.2 The 2017 Mainling earthquake-triggered landslides**
On November 18, 2017, a magnitude 6.4 earthquake struck Mainling County (95.02°E,
29.75°N), resulting in three injuries and affecting 12,000 individuals. The earthquake triggered
over 1,000 landslides, obstructing numerous watercourses and covering a total area of
33.61km², with the largest landslide spanning 4.9km² (Hu et al., 2019). Mainling County, located
on the southeastern margin of the Qinghai-Tibet Plateau within the Yarlung Zangbo Grand
Canyon, is part of the Mediterranean Himalayan seismic zone. This region, with altitudes
ranging from 800 to 7,782m and an average elevation of 2,500m, features a maximum elevation
differential of 2,000m and a robust vegetation coverage of 60% (Gao et al., 2023; Chen et al.,
2019). The monsoonal climate here brings annual rainfall between 1,500 and 2,000mm (Huang
et al., 2021). Following the earthquake, we acquired 3m-resolution PlanetScope images on
December 17, 2017, and April 08, 2018, to interpret the landslides (Figure.S2).
**3.1.3 The 2018 Hokkaido earthquake**
On September 6, 2018, a Mw 6.6 earthquake struck Hokkaido, Japan (142.01°E, 42.69°N),
resulting in 44 fatalities and over 660 injuries. Approximately 80% of the casualties were due to
coseismic landslides. The earthquake triggered over 7,800 landslides, causing extensive
damage to infrastructure. The total area affected by landslides was 23.77 km², with the largest
individual landslide covering 0.5km² (Wang et al., 2019). The region, which receives an annual
precipitation of 1,200–1,800mm—relatively low compared to other parts of Japan (Yamagishi
and Yamazaki, 2018)—is characterized by sandstone, mudstone, siltstone, and shale
formations, overlain by substantial volcanic sediments (Wang et al., 2019). Following the
Hokkaido earthquake, we acquired PlanetScope image with a 3m resolution on December 12,
2018, and Map World image with a 0.5m resolution (Figure.S3).

**3.1.4 The 2018 Palu earthquake**

On September 28, 2018, the Palu region of Sulawesi, Indonesia, was struck by a Mw 7.5
earthquake with a focal depth of 10 km (0.18°S, 119.84°E). A detailed analysis by Shao et al.
(2023) identified approximately 15,700 co-seismic landslides across a 14,600km² area, with a
combined landslide area of about 43.0km². These landslides were predominantly concentrated
in the mountainous canyon regions south of the epicenter. This study provides a semantic-level
interpretation of these landslides, which were mainly shallow disruptions (Shao et al., 2023).
However, some larger-scale flow slides, rockfalls, and debris flows were also observed. High-
resolution Map World image (1m) was utilized to support this analysis (Figure.S4).
earthquake

**3.1.5 The 2019 Mesetas earthquake**

The research site is located in the eastern foothills of the Colombian Eastern Cordillera.
On December 24, 2019, the Mesetas Earthquake, with a magnitude of 6.0, struck this region,
as documented by Poveda et al. (2022). The earthquake's epicenter was located at 76.19°W,
3.45°N, triggering approximately 800 co-seismic landslides. The distribution and predominant
orientation of these landslides were influenced by the shear zone confined within the Guapecito
Fault, a subsidiary offshoot of the Algeciras Fault (Poveda et al., 2022). High-resolution
PlanetScope images (3m) was acquired on January 5 and February 12, 2020, to analyze these
phenomena (Figure.S5).

**3.1.6 The 2021 Nippes earthquake**

On August 14, 2021, a Mw 7.2 earthquake struck the Nippes Mountains in Haiti (73.45°W,
18.35°N). This seismic event, compounded by heavy rainfall from Tropical Storm Grace on
August 16-17, triggered numerous secondary geological hazards across the Tiburon Peninsula.
The disaster resulted in at least 2,246 fatalities and injured over 12,763 individuals (Calais et
al., 2022). The earthquake-induced landslides totaled 4,893, covering an estimated 45.6km²,
with the largest individual landslide spanning $3.1×10^5$ m² (Zhao et al., 2022b). The affected area,
with elevations up to 2,300 m (Alpert, 1942), consists mainly of volcanic rocks, such as basalts,
and sedimentary formations, particularly limestones (Harp et al., 2016). Post-earthquake, we
utilized 3m-resolution PlanetScope image (August 29, 2022) and 0.5m-resolution Map World
image to assess the damage (Supplementary Figure 6).

On August 14, 2021, a seismic event registering Mw 7.2 hit in the Nippes Mountains of

Haiti (73.45°W, 18.35°N). This seismic activity, coupled with substantial rainfall from Tropical
Storm Grace between August 16 and 17, precipitated a significant number of secondary
geological hazards in the Tiburon Peninsula. The calamity resulted in a tragic loss of at least
2,246 lives and inflicted injuries upon more than 12,763 individuals (Calais et al., 2022). The
earthquake triggered a total of 4,893 landslides, covering an estimated area of 45.6km$^2$, with
the maximum individual area reaching $3.1×10^5$m$^2$ (Zhao et al., 2022b). The study area,
characterized by elevations reaching up to 2,300 m above sea level (Alpert, 1942). Comprised
predominantly of volcanic rocks, such as basalts, and sedimentary formations, notably
limestones (Harp et al., 2016). In addition to obtaining 3m-resolution PlanetScope image after
the Nippes earthquake, we also acquired 0.5m-resolution Map World image (Figure.S6).
**3.1.7 The 2022 Sumatra earthquake**

On February 25, 2022, a Mw 6.1 earthquake struck West Sumatra, Indonesia, at a shallow

depth of 4.9 km. The epicenter was located approximately 20 km from Mount Talakmau
(100.10°E, 0.22°N), a compound volcano rising to about 3,000m. Mount Talakmau, active
during the Holocene, consists of andesite and basalt from the Pleistocene-Holocene epoch
(Basofi et al., 2016). The earthquake induced extensive landslides over a 6km² area on the
volcano's eastern and northeastern flanks. High-resolution PlanetScope image (3m) from
March 5 and April 24, 2022, captured these landslides (Figure.S7).
**3.1.8 The 2022 Lushan earthquake**

On June 1, 2022, an Mw 5.9 earthquake (102.94°E, 30.37°N) struck Lushan County, China,

resulting in 4 fatalities and 42 injuries, affecting 14,427 individuals. The seismic event triggered
1,063 landslides over a total area of 7.2km², with the largest landslide covering 0.3km² (Zhao
et al., 2022a). This region, located on the southeast margin of the Qinghai-Tibet Plateau,
features an average elevation exceeding 2,000m, with altitudes ranging from 557 to 4,115m

(Tang et al., 2023). It is characterized by lush vegetation covering 80% of the area and experiences a subtropical monsoon climate with annual rainfall between 1,100 and 1,300mm (Chen et al., 2019). The geological composition predominantly consists of exposed sandstones and mudstones (Zhao et al., 2022a). High-resolution imagery, including 3 m-resolution PlanetScope images, 0.5m-resolution Map World image, and 0.2m-resolution UAV images acquired on June 13, 2022, using a Sony ILCE-5100, was collected for the affected region (Figure.S8).

**3.1.9 The 2022 Luding earthquake**

On September 5, 2022, an Mw 6.8 earthquake struck Luding County, China (102.08°E, 29.59°N), resulting in 93 fatalities. The seismic event triggered approximately 15,000 landslides over an area of 28.53km², with the largest individual landslide covering $2.4×10^5m^2$ (Dai et al., 2023). This region lies on the southeastern margin of the Qinghai-Tibet Plateau within the "Y"-shaped Xianshuihe Fault Zone (Yang et al., 2022b). The geological composition predominantly includes limestone, sandstone, dolomite, and some intrusive rocks (Dai et al., 2023). In the aftermath of the earthquake, rapid rescue operations and data collection were undertaken, utilizing 0.2m-resolution UAV image (acquired on October 7, 2022, via Phase One IXU1000), 3m-PlanetScope image (acquired on September 25, 2022), Map World image (0.5m), and Gaofen-6 (2m) (Figure.S9).

## 3.2 Preprocessing of landslide dataset

In the aforementioned nine events, the available public data primarily focuses on geological analysis rather than tasks related to semantic segmentation. After performing multi-source data spatial registration, atmospheric correction and radiometric calibration on remote sensing images, we used QGIS for landslide interpretation. These labels were delineated with reference to pre-earthquake remote sensing imagery and post-earthquake multi-source remote sensing image. By comparing spectral disparities and analyzing morphological attributes between bi-temporal images, we mapped the semantic landslide labels. (Figure.3). The mapping of landslide polygons for these nine events was primarily conducted by a team of five researchers,

including the authors. All team members possess expertise in geology or remote sensing and
were involved in a year-long process of detailed interpretation.

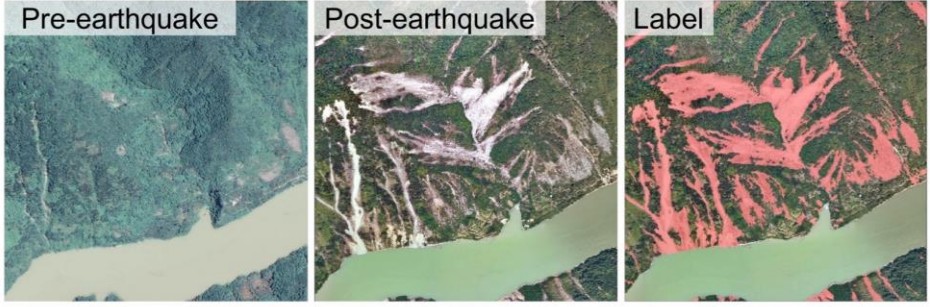

Luding earthquake region (UAV image)

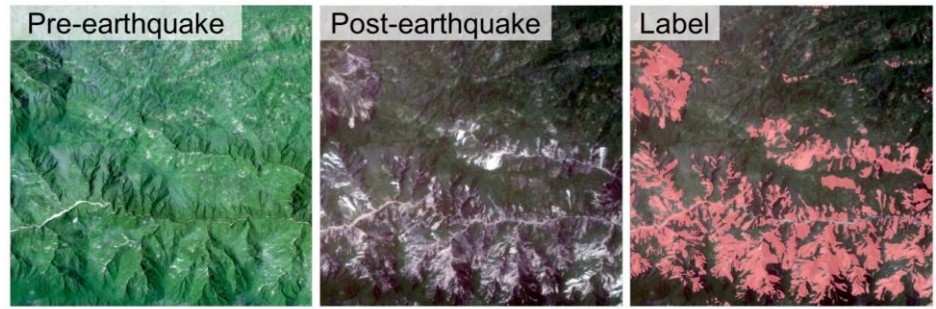

Haiti earthquake region (PlanetScope image)

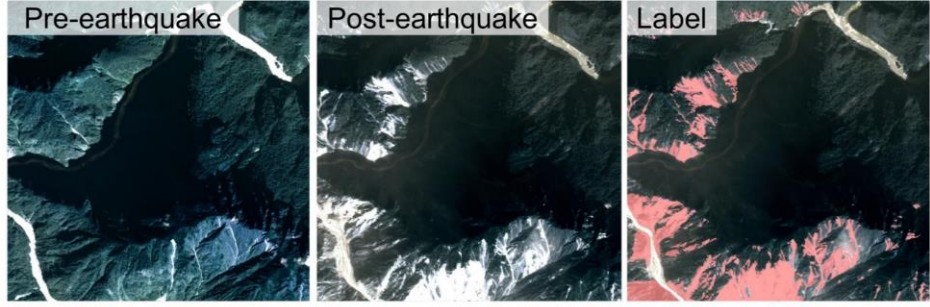

Mainling earthquake region (PlanetScope image)


**Figure.3** Remote sensing images before and after the earthquake and landslide interpretation
results (landslides marked in red).
Moreover, we actively participated in emergency response and field investigation after
these major earthquakes in China. This further improved the reliability of the landslide
inventories. Figure.4 showcases photographs captured on-site after the Jiuzhaigou earthquake,
Lushan earthquake, and Luding earthquake. Specifically, Figure.4 (A$_1$) and 4 (B$_1$) were taken
in Luding, Sichuan, depicting the extensive devastation caused by concentrated coseismic
landslides, impacting Wandonghe Village and resulting in severe destruction of local
infrastructure. Corresponding aerial photos with a resolution of 0.2m, Figure.4 ($A_2$) and 4 ($B_2$),
offer a comprehensive perspective of the affected area. Figure.4 ($C_1$), taken in Lushan, Sichuan,
captures the consequences of the earthquake-triggered large landslide dam, which obstructed
the river channel. The corresponding PlanetScope image, Figure.4 ($C_2$), provides an overhead
view of the altered landscape. Furthermore, Figure.4 ($D_1$), taken in the Jiuzhaigou Panda Sea,
illustrates a significant volume of landslide deposits reaching the sea, with the accompanying
UAV image at a resolution of 0.2m, Figure.4 ($D_2$), offering detailed insights. Lastly, Figure.4 (E)
presents a field work photo involved in these surveys. These field investigations serve to
enhance comprehension and subsequent calibration on our remote sensing interpretation.

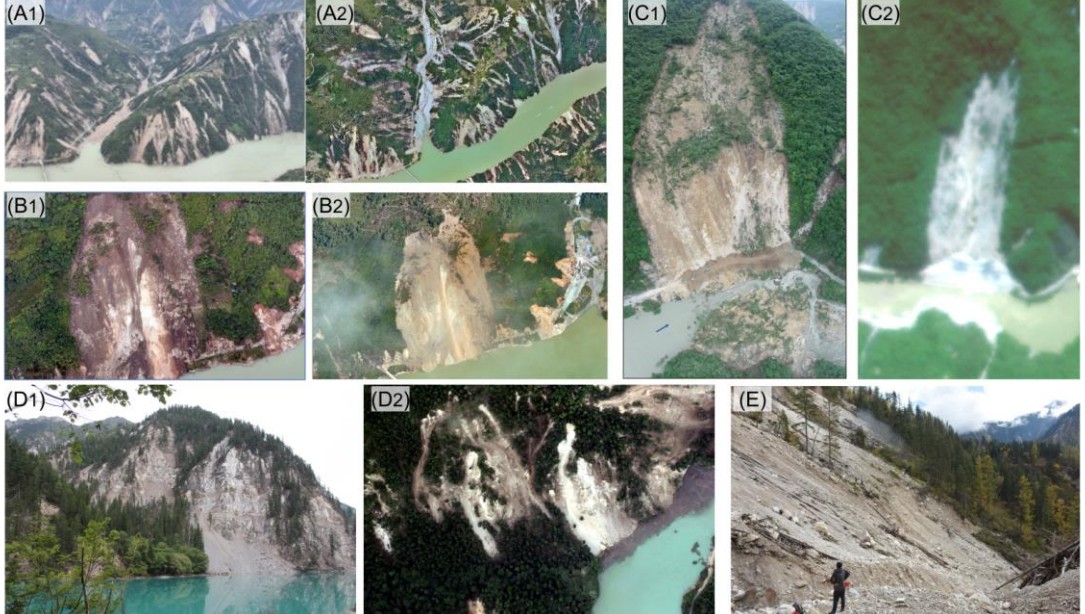

**Figure.4** Comparison of field survey photos and remote sensing images: $A_1$ and $A_2$ are the
Wandong landslides induced by the 2022 Luding earthquake; $B_1$ and $B_2$ are the Dadu River
Bridge landslide induced by the 2022 Luding earthquake; $C_1$ and $C_2$ are the Baoxing landslides
induce by the 2022 Lushan earthquake; $D_1$ and $D_2$ are the Panda sea landslides induced by
the 2017 Jiuzhaigou earthquake; E is a photo of field work at Jiuzhaigou.
To obtain semantic-level annotations for landslide labels, all remote sensing images were
converted into RGB images (8-bit). the preprocessing stage was conducted through three steps:
binary mask generation, data sampling, and image patching. First, utilizing the Rasterio library
in Python, landslide vector labels for each selected region were transformed into binary masks,

where 1 denoted landslide and 0 represented background. Subsequently, regions densely
populated with landslides were sampled, and both remote sensing images and masks were
patched and cropped into regular grids, yielding patches of 1,024×1,024 pixels. To mitigate
interference among patches, overlap parameter was set as 0. Given the obvious imbalance
between non-landslide and landslide areas, we manually removed most of the images without
any landslide pixel annotations. The ratios of positive landslide samples and negative non-
landslide samples were 8.01% and 91.99%, respectively. Table.3 presents detailed information
regarding different remote sensing data sources for each study case.

**Table.3** Detailed information of each event in GDCLD

| Events | Data sources | Resolution | Number of tiles |
|---|---|---|---|
| Jiuzhaigou 2017 (Mw 6.5) | UAV | 0.2m | 2,288 |
| | PlanetScope | 3m | 176 |
| Mainling 2017 (Mw 6.4) | PlanetScope | 3m | 118 |
| Hokkaido 2018 (Mw 6.6) | Map World | 0.5m | 796 |
| | PlanetScope | 3m | 123 |
| Palu 2018 (Mw 7.5) | Map World | 1m | 335 |
| Mesetas 2019 (Mw 6.0) | PlanetScope | 3m | 144 |
| Haiti 2021 (Mw 7.2) | PlanetScope | 3m | 238 |
| | Map World | 0.5m | 404 |
| Sumatra 2022 (Mw 6.1) | PlanetScope | 3m | 110 |
| Lushan 2022 (Mw 5.9) | UAV | 0.2m | 210 |
| | Map World | 0.5m | 182 |
| | PlanetScope | 3m | 110 |
| Luding 2022 | UAV | 0.2m | 9,252 |

| | | | |
|---|---|---|---|
| (Mw 6.6) | Map World | 0.5m | 1,540 |
| | GF-6 | 2m | 496 |
| | PlanetScope | 3m | 190 |
| Sum | - | - | 16712 |

Additionally, to enhance the robustness and generalization capability of deep learning
models, a subset of background noise elements such as clouds, roads, buildings, bare land,
and rocks were manually selected as negative non-landslide samples. The negative samples
can be outlined as follows: diverse roads (Figure.5: (e), (k), (m), (n), (p), (s)), river channels
(Figure.5: (e), (k), (n), (s), (t)), clouds (Figure.5: (o), (r)), barren land (Figure.5: (c), (h), (q)).
Additionally, human-engineered structures and buildings are also considered (Figure.5: (e), (k)).

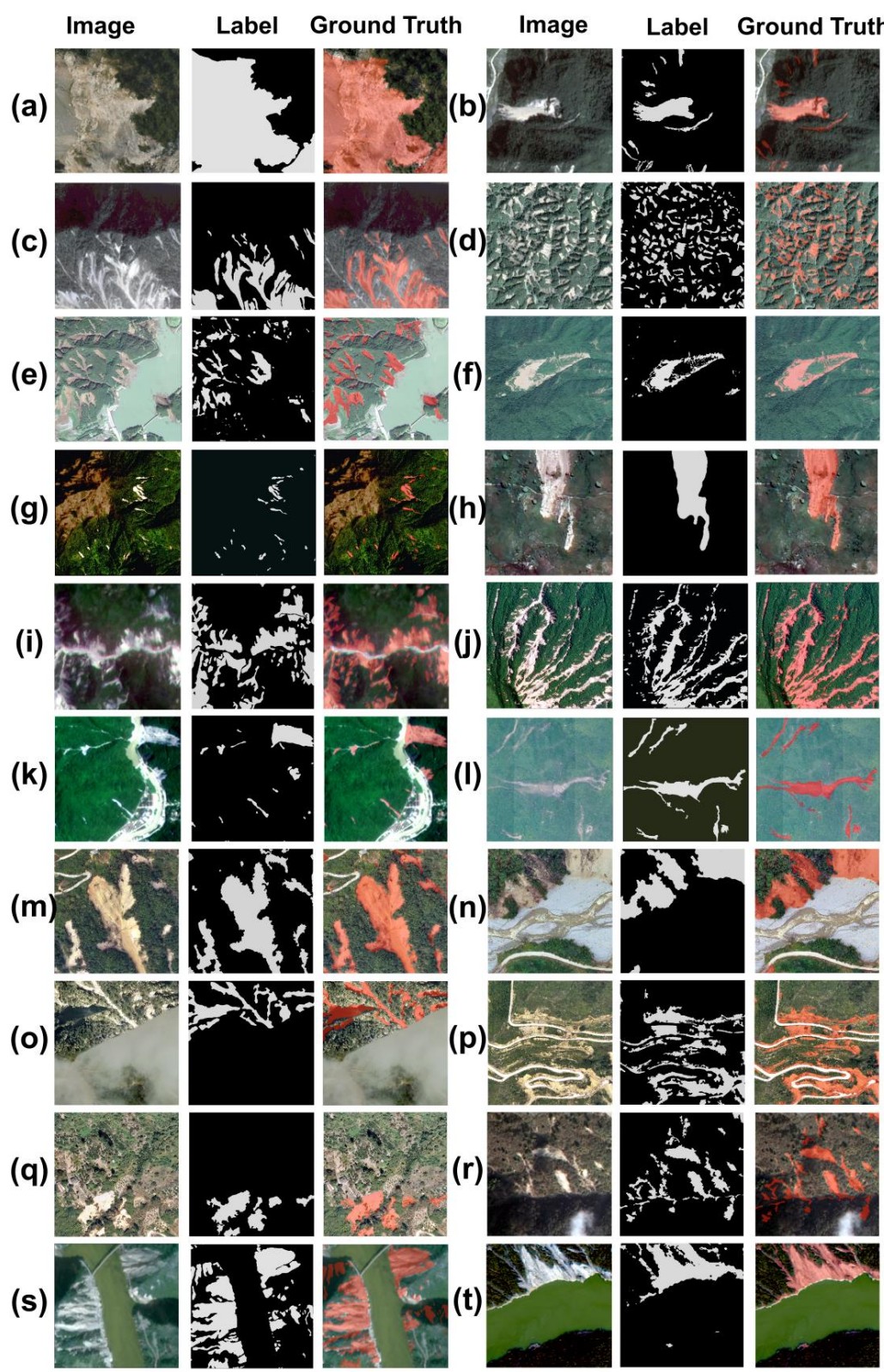

Figure.5 Display of landslide sample data from different study areas and different remote sensing data sources: Jiuzhaigou UAV (a), Jiuzhaigou PlanetScope (b), Mainling PlanetScope (c), Hokkaido PlanetScope(d), Hokkaido Map World (e), Palu Map World (f), Mesetas

PlanetScope (g), Haiti Map World (h), Haiti PlanetScope (i), Sumatra PlanetScope (j), Lushan
PlanetScope (k), Lushan UAV (l), Luding UAV(m~q), Luding Map World (r), Luding PlanetScope
(s), and Luding Gaofen-6 (t). The "label" refers to the binary landslide mask, whereas the
"Ground Truth" illustrates the concordance between the annotated and actual landslide in
images.

# 354    4. Experimental setup

After the completion of dataset construction, the experimental phase follows. In this section,

we will introduce several semantic segmentation algorithms used for validating the dataset, the
loss functions and accuracy evaluation metrics employed in the experiments, as well as various
hyperparameter settings utilized during the experiments.

## 359    4.1 Segmentation algorithms

In this section, we have selected seven of the most popular semantic segmentation

networks, including four models based on the CNN architecture and three based on the
Transformer architecture. These seven algorithms have medium to large-scale parameter sizes
and computational complexities, and show excellent performance in a variety of remote sensing
semantic scenarios, making them suitable for Precision comparison and validation of novel
datasets.

(1) UNet: As one of the earliest and most renowned semantic segmentation models, UNet

is distinguished by its unique U-shaped architecture (Ronneberger et al., 2015). This design
facilitates efficient learning and precise localization by combining high-resolution features from
the contracting path with up-sampled outputs from the expanding path. Both the encoder and
decoder in UNet are composed purely of CNN structures (O'shea and Nash, 2015). This
simplicity, along with a relatively small number of parameters, allows UNet to achieve
exceptional accuracy and rapid inference on small datasets. Consequently, it is widely utilized
in applications such as small-scale object classification, change detection, and medical imaging.

(2) ResUNet: ResUNet is an advanced variant of the UNet model, incorporating residual

connections to enhance its performance and learning efficiency (Diakogiannis et al., 2020). The
key innovation in ResUNet is the integration of residual blocks within both the encoder and
decoder paths, which address the vanishing gradient problem and enable the training of deeper
networks (He et al., 2016). These residual blocks allow the network to learn identity mappings,
facilitating gradient flow through the network and improving convergence rates. Similar to UNet,
ResUNet maintains a U-shaped architecture that combines high-resolution features from the
contracting path with up-sampled outputs from the expanding path, ensuring precise
localization and context capture. The combination of residual connections improves feature
reuse and learning efficiency, enabling ResUNet to effectively improve Recall and small target
detection capabilities in semantic segmentation tasks.
(3) DeepLabV3: DeepLabV3, is a semantic segmentation model known for its
sophisticated use of atrous convolution, or dilated convolution (Chen et al. 2018). This
technique allows the network to capture multi-scale contextual information without losing spatial
resolution, addressing the limitations of traditional convolutional networks in dense prediction
tasks. DeepLabV3 incorporates atrous spatial pyramid pooling to robustly segment objects at
multiple scales by applying atrous convolution with different rates in parallel. This model also
integrates features from both the encoder and decoder paths, enhancing the Precision of
boundary delineation. In addition, the architecture of DeepLabV3 utilizes batch normalization
and depth-separable convolution. This design can effectively reduce the complexity and
computational cost of the model, while enabling the model to have stronger feature extraction
capabilities and generalization than simple networks such as UNet.
(4) HRNet: High-Resolution Network (HRNet) is noted for its innovative approach to
maintaining high-resolution representations throughout the network (Wang et al., 2020). Unlike
traditional models that gradually down-sample the input to extract features, HRNet preserves
high-resolution features by maintaining parallel high-to-low resolution subnetworks. This design
allows HRNet to integrate multi-scale information effectively, ensuring precise localization and
robust feature representation. The network continuously exchanges information across
different resolutions, resulting in superior accuracy and detailed segmentation results. Due to
its ability to retain fine-grained spatial information and adapt to various scales, HRNet excels in
complex tasks such as fine-grained terrain classification, semantic segmentation in urban
scenes, and fine-grained visual detection.
(5) UperNet: UperNet employs a pyramid feature extraction method, integrating multi-scale
information to capture contextual details across different resolutions (Xiao et al., 2018; Liu et
al., 2022). It utilizes a Feature Pyramid Network (FPN) backbone for hierarchical feature
extraction, enhanced by a global context integration module to enrich overall scene
understanding. Additionally, UperNet incorporates lateral connections for efficient
communication between feature pyramid levels, ensuring seamless information flow and
accurate segmentation. This sophisticated architecture enables UperNet to achieve superior
segmentation performance, particularly in challenging scenarios with complex scenes and
diverse object scales.
(6) SwinUNet: Built upon the Swin Transformer architecture, SwinUNet blends self-
attention mechanisms with UNet for exceptional performance (Cao et al., 2022). It inherits Swin
Transformer's hierarchical feature extraction for capturing both local and global contextual
information efficiently (Liu et al., 2021). The self-attention mechanism enables capturing
nuanced relationships in data. SwinUNet integrates UNet's contracting and expanding paths in
decoding, emphasizing spatial detail preservation. This combination empowers SwinUNet to
excel in tasks requiring precise localization and robust contextual understanding. (7)
SegFormer: SegFormer, represents a significant advancement in semantic segmentation by
leveraging a transformer-based architecture (Xie et al., 2021). Unlike traditional CNN
approaches, SegFormer employs a hierarchical transformer encoder to capture multi-scale
contextual information effectively, without relying on complex designs such as positional
encodings or large pre-training datasets. The decoder in SegFormer integrates features from
different scales using lightweight multi-layer perceptron, ensuring efficient and precise
segmentation. This innovative design enables SegFormer to achieve excellent segmentation
results with medium-sized parameters and fast inference speed in high-resolution complex
scenes.

## 4.2 Loss function and accuracy evaluation

Since the landslide detection is a two-class semantic segmentation task, we choose the Binary Cross-Entropy (De Boer et al., 2005) as the loss function for model training, whose mathematical expression is shown as follow:

$$L(y,\hat{y})=-\frac{1}{N}\sum_{i=1}^{N} [y_i \log(\hat{y}_i) +(1-y_i)\log(1-\hat{y}_i)] \tag{1}$$

where L is the loss function, N is the number of samples, $y_i$ is the true label (0 or 1) of the i-th sample, and $\hat{y}_i$ is the predicted probability of the i-th sample.

For accuracy evaluation, the following accuracy indicators are calculated through confusion matrices (Townsend, 1971): Precision, Recall, F1 score (Chicco and Jurman, 2020) and mean intersection over union (mIoU) (Rezatofighi et al., 2019). Their calculation formulas are as follows:

$$Precision=\frac{TP}{TP+FP} \tag{2}$$

$$Recall=\frac{TP}{TP+FN} \tag{3}$$

$$F1=\frac{2\times Precision\times Recall}{Precision+Recall} \tag{4}$$

$$mIoU=\frac{1}{N}\sum_{i=1}^{N} \frac{TP_i}{TP_i+FP_i+FN_i} \tag{5}$$

where the TP is the True Positive, FP is the False Positive, TN is the True Negative and FN is the False Negative.

## 4.3 Equipment and Parameter

The deep learning framework employed in this study is conducted based on PaddlePaddle 2.3.2 (Ma et al., 2019), with the environment configured for Python 3.8, CUDA 11.2, and CuDNN 8.3.0. The experimental setup encompasses Intel Xeon CPU, W2255, 3.7GHz, equipped with 256GB of system memory. The GPU infrastructure consists of Tesla V100, with 32GB of video memory. The operating system employed is Ubuntu 20.04. The model's optimizer is selected as AdamW (Loshchilov and Hutter, 2017), with an initial learning rate of 0.0006, beta1 set to 0.9, beta2 to 0.999, weight decay to 0.01 and epoch to 100.

# 5. Results

To validate the accuracy of the GDCLD dataset, this study selected four types of remote sensing images (UAV, PlanetScope, Map World image, and Gaofen-6) from five seismic events (Luding, Jiuzhaigou, Hokkaido, Mainling, and Nippes) as training and validation datasets for model construction and accuracy evaluation. The ratio of training dataset to validation dataset is 3:1. To further assess the generalization ability of this dataset, we chose three types of remote sensing images (UAV, PlanetScope, and Map World image) from four independent seismic events (Lushan, Mestas, Sumatra, and Palu) as the test dataset. Considering the geographical distribution, these four regions, located on different continents and characterized by distinct tectonic settings and climatic conditions, ensure complete independence from the training dataset. From the perspective of data sources, the four study areas represent three major types of remote sensing imagery: PlanetScope, UAV, and Map World. Additionally, the UAV sensor used in the Lushan area is different from those used in other regions. This data partitioning strategy is designed to rigorously evaluate the generalization capability of the GDCLD-trained model.

We conducted evaluations on our dataset utilizing the aforementioned seven semantic segmentation algorithms. After each model is trained for 100 epochs, we meticulously examined the performance of the GDCLD dataset in landslide identification. we present the performance of the seven algorithms on the validation dataset in Table.4.

Among these seven algorithms, UNet, ResUNet, DeepLabV3, and HRNet serve as neural network models with convolutional structures, whereas UperNet, SwinUNet, and SegFormer are based on transformer-based neural network architectures. From Table.4, it is evident that Transformer-based semantic segmentation models exhibit superior performance compared to models based on convolutional structures. Overall, the mIoU of the seven algorithms on GDCLD validation set spans from 71.07% to 85.06%. Notably, UNet demonstrates the least detection with the mIoU and F1 score of 71.07% and 79.54%. In contrast, SegFormer yields the best performance with the accuracy of 91.35%, Recall of 91.70%, F1 score of 91.52%, and

mIoU of 85.06%. Figure.6 illustrates the detection results of different models across various
remote sensing data sources. it can be seen that transformer-based semantic segmentation
models achieve superior segmentation outcomes.

**Table.4** Comparison of result on GDCLD validation dataset

| Method | Backbone | Precision (%) | Recall (%) | F1 (%) | mIoU (%) |
|---|---|---|---|---|---|
| UNet | - | 77.05 | 82.01 | 79.54 | 71.07 |
| ResUNet | ResNet-50 | 78.17 | 86.48 | 82.11 | 71.94 |
| DeepLabV3 | ResNet-50 | 81.27 | 86.96 | 84.02 | 74.61 |
| HRNet | HRNet-48 | 81.88 | 87.21 | 84.46 | 75.19 |
| UperNet | ViT-B16 | 88.18 | 90.64 | 89.39 | 81.97 |
| SwinUNet | - | 89.78 | **92.01** | 90.72 | 83.68 |
| SegFormer | MiT-B4 | **91.35** | 91.70 | **91.52** | **85.06** |

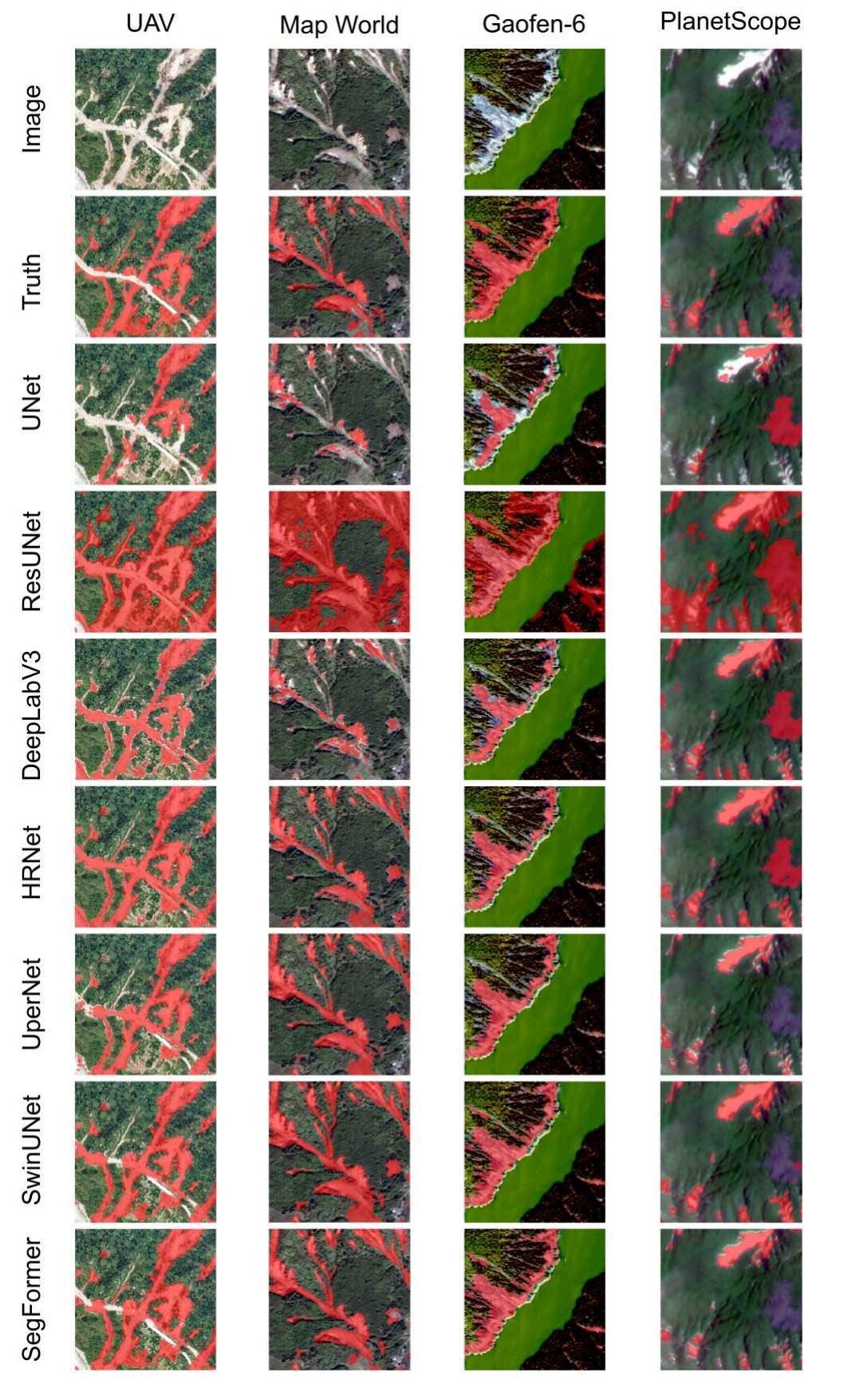


**Figure.6** Comparative results of different algorithms on validation dataset

To demonstrate the robustness and generalization capability of the dataset-trained models
in other environment, we conducted testing by using four independent events, as illustrated in
Table.5. Overall, the mIoU performance of the algorithms trained on GDCLD ranges from 56.09%
to 72.84%. SegFormer exhibits the best performance, achieving Precision of 77.09%, Recall of
87.09%, F1 score of 81.88%, and mIoU of 72.84%. we present detailed results of six types of
remote sensing images in these four events in Table.6. The overall mIoU ranges from 69.01%
to 82.31%, while the F1 ranges from 80.63% to 89.30%. Furthermore, we noticed a remarkable
imbalance between Recall and Precision in the predicted results. The Recall is always higher
than the Precision, as it is crucial to not miss any important landslides for disaster assessment
and rescue operations. From the perspective of remote sensing sensors, except for the
Sumatra incident, higher resolution was directly related to better landslide detection
performance.
**Table.5** Comparison of result on test dataset

| Method | Backbone | Precision (%) | Recall (%) | F1 (%) | mIoU (%) |
|--------|----------|---------------|------------|--------|----------|
| UNet | - | 61.69 | 61.22 | 61.45 | 56.09 |
| ResUNet | ResNet-50 | 66.56 | 64.46 | 65.49 | 57.06 |
| DeepLabV3 | ResNet-50 | 65.26 | 67.75 | 66.48 | 59.73 |
| HRNet | HRNet-48 | 65.52 | 72.03 | 68.62 | 61.79 |
| UperNet | ViT-B16 | 69.96 | 78.08 | 73.80 | 65.42 |
| SwinUNet | - | 71.56 | 82.26 | 76.54 | 67.18 |
| SegFormer | MiT-B4 | **77.09** | **87.09** | **81.88** | **72.84** |

**Table.6** Detection results of SegFormer in different events

| Events | Image type | Precision (%) | Recall (%) | F1 (%) | mIoU (%) |
|--------|-----------|---------------|------------|--------|----------|
| Lushan | UAV | 74.72 | 90.35 | 81.80 | 72.96 |
|  | Map World | 76.18 | 87.35 | 81.38 | 71.92 |
|  | PlanetScope | 81.50 | 82.28 | 81.78 | 69.05 |
| Palu | Map World | 73.48 | 91.24 | 81.40 | 71.12 |
| Mesetas | PlanetScope | 80.26 | 80.97 | 80.63 | 69.01 |
| Sumatra | PlanetScope | 83.57 | 97.45 | 89.30 | 82.31 |

Figures.7 to 10 respectively illustrate the detection results for Mesetas (PlanetScope),
Sumatra (PlanetScope), Palu (Map World image), and Lushan (UAV). The F1 score of the
Mesetas event model is 80.63%, with Recall and Precision exhibiting relative balance. As
observed in Figure.7, our model demonstrates strong capabilities in detecting and segmenting
the majority of landslides, particularly in regions of mountainous slopes (Figure.7 (d)). In areas
affected by mountain shadows (Figure.7 (b, c, e)), as expected, since, pixel signatures of
shadows are very different than those of landslides. The model effectively identifies most large
landslides but exhibits some omissions in detecting small landslides. In the Sumatra event, we
attained remarkably excellent detection results, with F1 score of 89.30%, Recall of 97.45%, and
Precision of 83.57%, Recall is 13.88% higher than Precision. As illustrated in Figure.8, the
model effectively identifies nearly all landslides (Figure.8 (b, c)). However, there are instances
of missed landslide detection in the lower-right corner of Figure.8 (a). This is due to the apparent
confusion between the landslide accumulation area and river channels, resulting in sub-optimal
detection. In the Palu event, our F1 score yielded a result of 81.40%, with Recall reaching 91.24%
and Precision by 73.48%, Recall is 17.76% higher than Precision. As depicted in Figure.9, the
detection outcomes effectively discriminate between numerous cloud obscuration, bare lands,
and buildings, underscoring the positive efficacy of augmenting negative samples in our dataset
to improve the model's detection capabilities. Similarly, for the Lushan event captured by UAV,
we achieved the F1 score of 81.80%, with Recall and Precision of 90.35% and 74.72%, Recall
exceeding Precision by 15.63%. As shown in Figure.10, in the UAV data, the model
demonstrates exceptional segmentation capabilities for large-scale landslides (Figure.10 (b, c,
d)), while its detection performance for some small-scale disasters is less satisfactory. Overall,
the model trained based on GDCLD demonstrated excellent generalization capabilities across
four independent test datasets. It successfully detected all major landslides and effectively
segmented landslide boundaries. More importantly, the model effectively excluded background
noise from river channels, bare ground in residential areas, and cloud region, showcasing its
remarkable robustness.

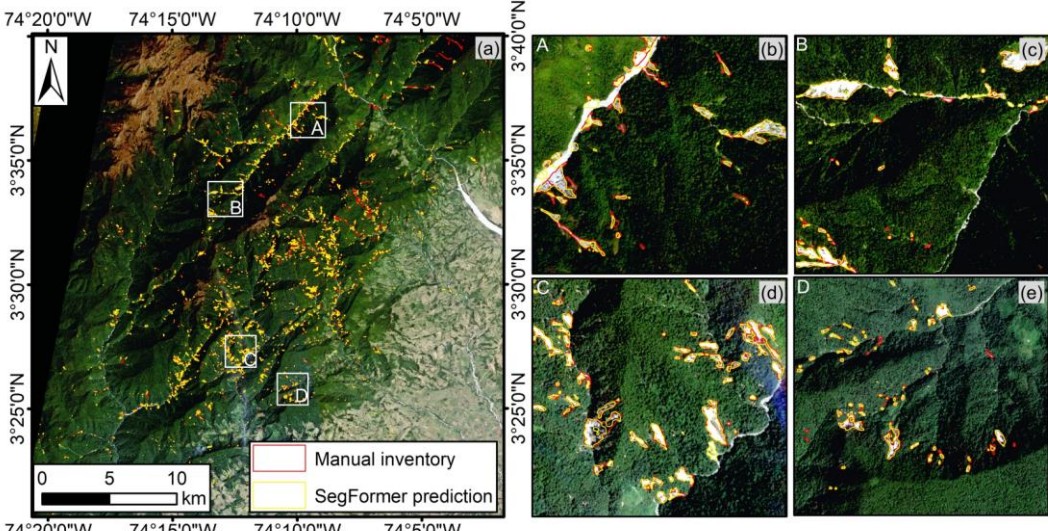


**Figure.7** Mesetas PlanetScope dataset. (a) Regional aerial view. (b-e) Detection results of four
magnified areas.

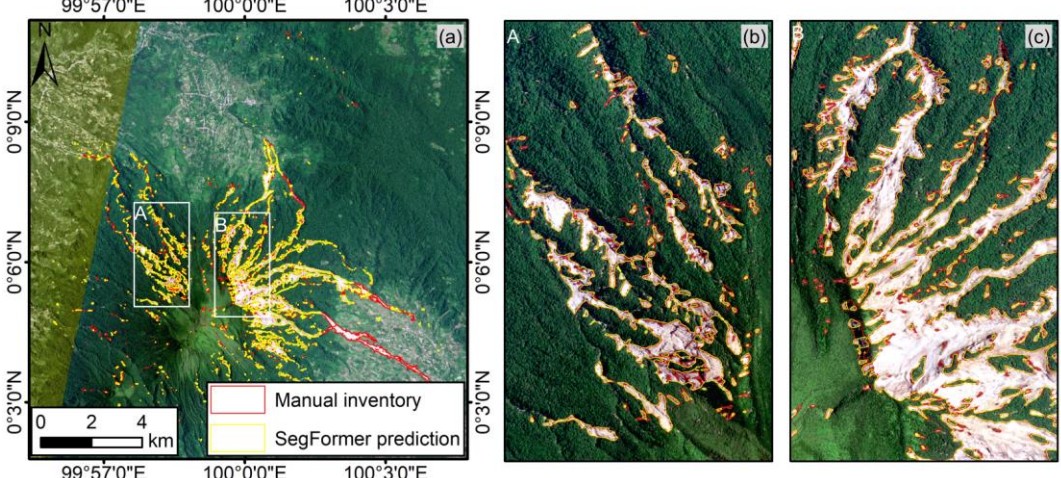


**Figure.8** Sumatra PlanetScope dataset. (a) Regional aerial view. (b-c) Detection results of two
magnified areas.

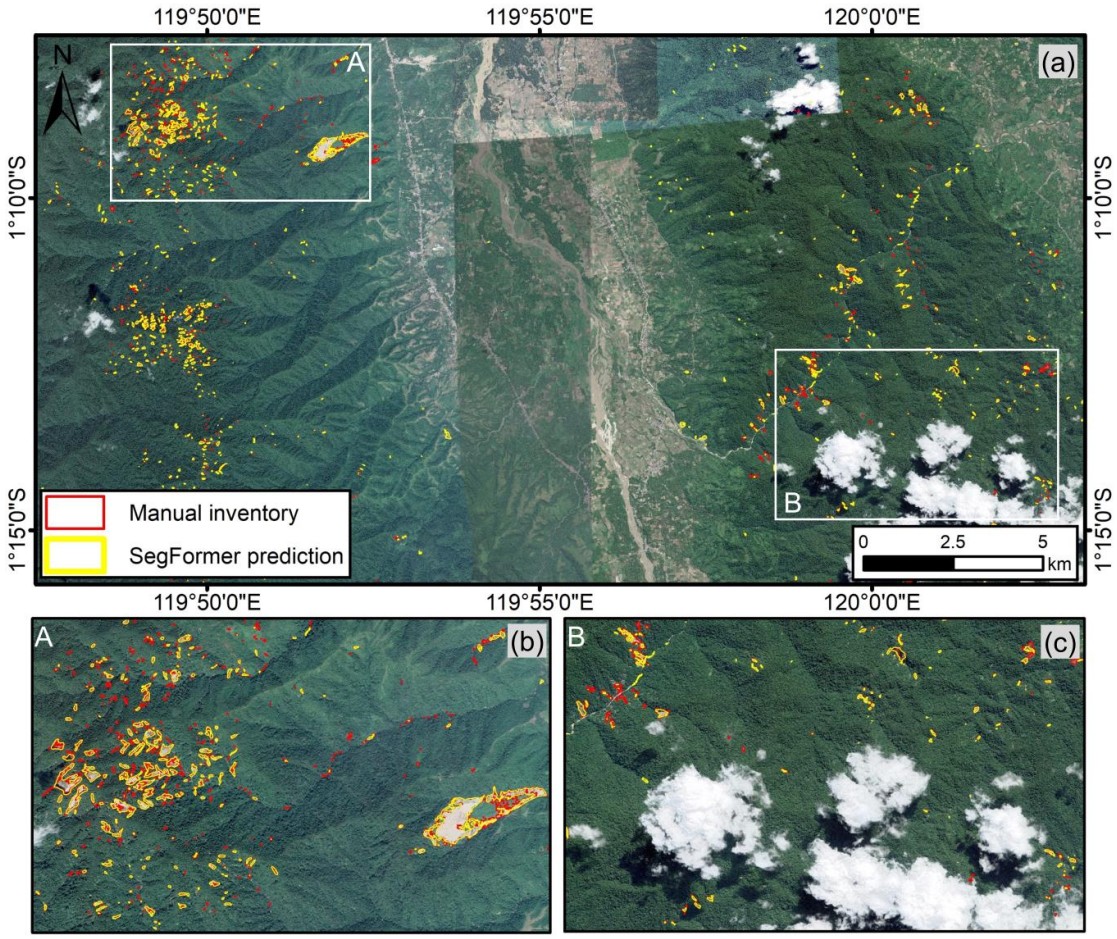


**Figure.9** Palu Map World dataset. (a) Regional aerial view. (b-c) Detection results of two
magnified areas.

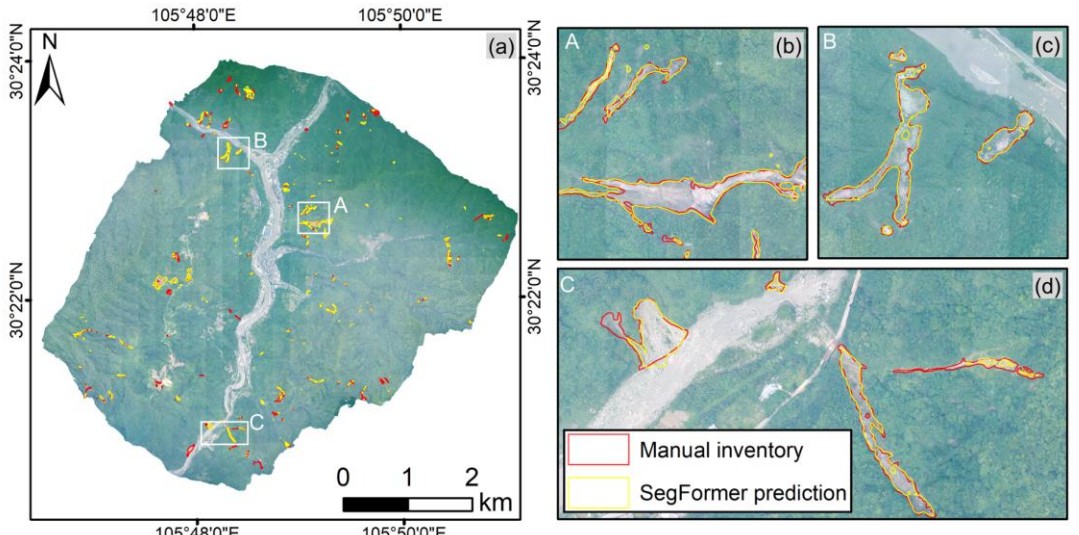


**Figure.10** Lushan UAV dataset. (a) Regional aerial view. (b-d) Detection results of three
magnified areas.

# 6. Discussion

## 6.1 Sample richness of GDCLD

The GDCLD dataset stands out as the most extensive and comprehensive repository of landslide data currently available, encompassing landslide data from various geographic environments and multiple remote sensing sources. the annotated landslide labels within this dataset tally up to approximately $1.39 \times 10^9$ pixels, roughly six times as many annotations as all the other publicly accessible landslide datasets (Figure.11). Additionally, this dataset includes a variety of negative samples with optical characteristics similar to landslides which can significantly enhance the model's generalization capability. In contrast to other datasets, which are limited to training small-scale semantic segmentation models like UNet and DeepLabV3 (Xu et al., 2024; Meena et al., 2022; Ghorbanzadeh et al., 2022), the GDCLD dataset can effectively train large-scale semantic segmentation models such as Transformers. Moreover, unlike Sentinel-2 and Landsat satellite image, where moderate spatial resolutions can limit the accurate delineation of landslide boundaries, GDCLD provides remarkably high spatial resolutions (0.2m~3m) and diverse spectral characteristics. This dataset not only performs well in landslide mapping across diverse geographical settings, but also serves as a baseline dataset for transfer learning in landslide detection.

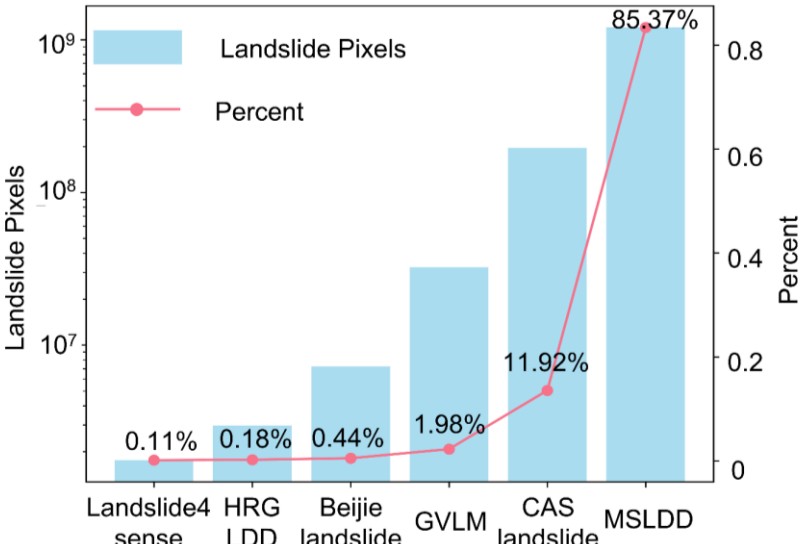

**Figure.11** Statistical comparison of landslide pixels in different landslide datasets.

## 6.2 Enhancement in model generalization

In the GDCLD dataset, a general selection of remote sensing data from multiple sources enhances the overall generalization capability of the landslide identification model. To substantiate this assertion, we conduct a comparative analysis between models trained by single- and multi-source datasets. The datasets from different sensors are segregated, and the SegFormer, which is an advanced and widely used transformer-based algorithm, is applied to train the landslide models. Their performance was verified by their respective test dataset as well as an independent event of Lushan earthquake.

The accuracy metrics for the validation dataset are presented in Table.7. Across four remote sensing sources—PlanetScope, Gaofen-6, Map World, and UAV—models trained on single-source datasets consistently demonstrate higher performance on test samples, with mIoU indices surpassing those of multi-source datasets by 2.26%, 1.63%, 0.64%, and 0.13%, respectively. However, a noteworthy observation emerges when models are transferred to the independent Lushan earthquake case (Table.8). The model trained on the multi-source dataset achieves significantly enhanced performance compared to the model derived from single-source counterpart. The mIoU of UAV-, Map World- and PlanetScope based datasets are improved by 8.16%, 7.95% and 0.09%. As depicted in Figure.12, the models trained by multi-source images exhibit higher recalls, accurate landslide boundaries, and robust resistance to interference. The yellow circle highlights the enhancements of models trained by multi-source images compared to single-source images. From the perspective of data sources, Map World contains different types of images (such as Gaofen and Jilin), encompassing multitude of spectral responses across these sensors. the UAV image in the Lushan event utilize the sensor different from those in the Luding and Jiuzhaigou event, resulting in noticeable spectrum differences in images. Consequently, compared to a single remote sensing source, the generalization capability of the models trained by multi-source images demonstrate a more pronounced improvement. In contrast, the PlanetScope image, obtaining from the same satellite sensors, exhibits smaller spectral variations in various images. As a result, the model trained on both single and multi-source datasets achieve similar performance. This highlights

the importance of datasets with diverse images sources for enhanced model performance in
landslide mapping. This indicate that the utilization of multi-source remote sensing datasets
enables the model to learn the spectral characteristics of the images from diverse sensors.
Hence, the model trained by GDCLD possesses enhanced generalization ability and
robustness, enabling it to effectively perform landslide mapping in independent cases without
prior knowledge.
**Table.7** GDCLD performances on validation dataset through single- and multi-source dataset

| Data source | Data type | Precision (%) | Recall (%) | F1 (%) | mIoU (%) |
|---|---|---|---|---|---|
| Single source | UAV | **92.20** | **92.90** | **92.54** | **87.07** |
| | PlanetScope | **87.98** | **87.81** | **87.89** | **80.11** |
| | Map World | **86.49** | **90.01** | **88.21** | **80.66** |
| | Gaofen-6 | **91.25** | **88.04** | **89.62** | **83.61** |
| Multiple source | UAV | 91.91 | 92.64 | 92.27 | 86.94 |
| | PlanetScope | 85.01 | 87.79 | 86.37 | 77.85 |
| | Map World | 86.42 | 89.12 | 87.74 | 80.02 |
| | Gaofen-6 | 90.49 | 85.20 | 87.77 | 81.98 |

**Table.8** GDCLD performances on unseen dataset through single- and multi-source dataset

| Data source | Data type | Precision (%) | Recall (%) | F1 (%) | mIoU (%) |
|---|---|---|---|---|---|
| Single source | UAV | 64.92 | **90.68** | 75.67 | 64.80 |
| | PlanetScope | 81.25 | **82.29** | 81.75 | 68.96 |
| | Map World | 68.39 | 80.16 | 73.81 | 63.97 |
| Multiple source | UAV | **74.72** | 90.35 | **81.80** | **72.96** |
| | PlanetScope | **81.50** | 82.28 | **81.78** | **69.05** |
| | Map World | **76.18** | **87.35** | **81.38** | **71.92** |

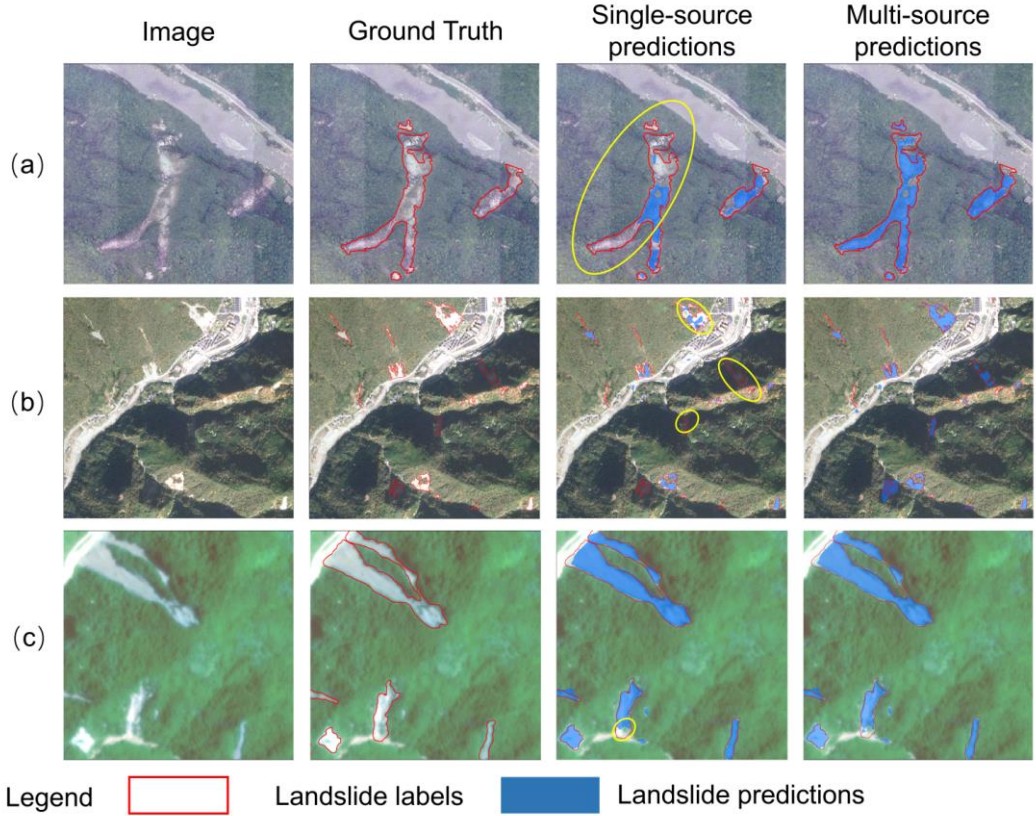

**Figure.12** Comparative results of ablation experiments between multi- and single-source (a).

UAV, (b). Map World, (c). PlanetScope

## 6.3 Comparison with existing landslide datasets and models

To assess the robustness and generalization capabilities of the GDCLD dataset, we

employ SegFormer trained on the GDCLD dataset (GDCLD-S model) to identify landslides

within three distinct datasets: HR-GLDD, GVLM, and CAS. Initially, we standardize the data

from these three datasets into 1024×1024 remote sensing tiles. Subsequently, utilizing the

GDCLD-S model, we conduct landslide identification across all these datasets. Table.9

demonstrates favorable performance of the model across these diverse datasets. For instance,

in the HR-GLDD dataset, which shares similarities with the PlanetScope image in GDCLD, the

model achieves an mIoU of 76.97%, indicating a balance between Precision and Recall metrics.

Similarly, when applied to the GVLM dataset, leveraging Map World image, our dataset exhibits

robust predictive outcomes, resulting in a comprehensive mIoU of 70.07%. Likewise, for the

CAS dataset, GDCLD demonstrates strong generalization capabilities, yielding an outstanding

comprehensive metric with mIoU = 76.91%, alongside balanced Recall and Precision metrics.

Although all landslide samples contained in GDCLD are induced by seismic activity, our model demonstrates good detection capabilities for rainfall-induced landslides. These two categories exhibit distinct spectral characteristics from their surrounding environments. Consequently, models trained on GDCLD exhibit proficient detection capabilities for rainfall-induced landslides. We present the identification performance of GDCLD-based model for rainfall-induced landslides from the GVLM dataset in Table.9 and Figure.13. Figure.13 underscores the excellent detection performance of the GDCLD-S model on rainfall-induced landslides in the GVLM dataset. Despite occasional misclassifications of small-size targets, the model effectively delineates the majority of rain-induced landslides. the Precision metrics in Table.8 affirm this observation with an mIoU reaching 78.22% and both Recall and Precision exceeding 85%. This highlights the robust generalization capability of the model trained by our dataset, enabling effective identification of rainfall-induced landslides.

**Table.9** Validation results of other public datasets

| Dataset | Precision (%) | Recall (%) | F1 (%) | mIoU (%) |
|---------|---------------|------------|--------|----------|
| HR-GLDD | 84.88 | 86.81 | 85.84 | 76.97 |
| GVLM | 72.83 | 87.54 | 80.68 | 70.07 |
| CAS | 82.95 | 86.35 | 84.62 | 76.91 |
| GVLM-rainfall | 85.88 | 86.71 | 86.29 | 78.22 |

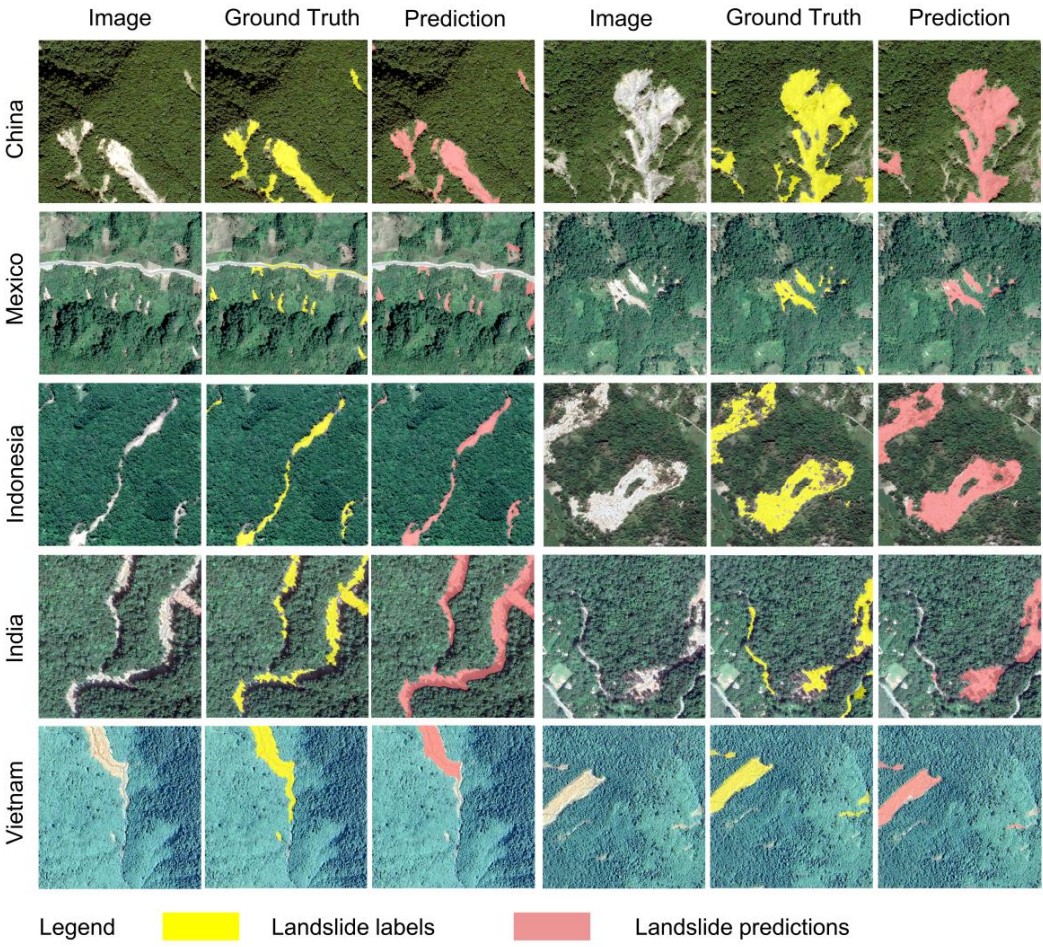

| Image | Ground Truth | Prediction | Image | Ground Truth | Prediction |

Legend   ■ Landslide labels   ■ Landslide predictions

**Figure.13** Detection results of rainfall landslides by GDCLD-S model. Map credits: GVLM.

In addition to the aforementioned analyses, we compare the performance of GDCLD with other two datasets, GVLM and CAS. Specifically, we train landslide detection models using the SegFormer algorithm on the GVLM and CAS datasets, denoted as GVLM-S and CAS-S, respectively, with identical training parameters as previously described. Furthermore, we also use the DeepLabV3 to train the CAS-D model based on the CAS dataset and use it for comparison of landslide detection (Xu et al., 2024). Subsequently, the GDCLD-S, CAS-S, CAS-D and GVLM-S models were applied to identify landslides in the Lushan area using three distinct remote sensing data sources: UAV, PlanetScope, and Map World. The results of this comparison are presented in Table 10. From Table 10, it is evident that the GDCLD-S model outperformed CAS-S, CAS-D and GVLM-S across all three remote sensing datasets, achieving mIoU of 72.96%, 69.05%, and 71.92% on UAV, PlanetScope, and Map World. In contrast, CAS-S records mIoU values of 62.03%, 56.86%, and 60.35% for the same datasets, respectively,

which is better than the CAS-D model trained with DeepLabV3, and also illustrates the advantages of the transformer architecture over the CNN architecture. Notably, GDCLD-S exhibited a significantly higher Recall than the other two models and also demonstrated an advantage in Precision. Overall, GDCLD-S, along with CAS-S, exhibited superior performance compared to the single-source data model GVLM-S, particularly in handling multisource remote sensing images. The extensive landslide data and negative samples included in GDCLD-S further contributed to its enhanced robustness against noise and improved Recall in landslide detection.

**Table.10** Performance comparison of GDCLD-S, GVLM-S, CAS-S, CAS-D on the Lushan dataset

| Model | Data type | Precision (%) | Recall (%) | F1 (%) | mIoU (%) |
| --- | --- | --- | --- | --- | --- |
| CAS-D | UAV | 72.73 | 55.34 | 62.88 | 57.91 |
| | PlanetScope | 52.07 | 56.05 | 53.93 | 52.86 |
| | Map World | 61.79 | 70.50 | 64.9 | 58.11 |
| GVLM-S | UAV | 73.03 | 54.84 | 57.67 | 53.41 |
| | PlanetScope | 60.13 | 53.40 | 54.82 | 51.52 |
| | Map World | **77.71** | 66.40 | 71.56 | 63.97 |
| CAS-S | UAV | 74.08 | 67.05 | 69.95 | 62.03 |
| | PlanetScope | 58.56 | 76.57 | 66.40 | 56.86 |
| | Map World | 75.02 | 64.65 | 68.37 | 60.35 |
| GDCLD-S | UAV | **74.72** | **90.35** | **81.80** | **72.96** |
| | PlanetScope | **81.50** | **82.28** | **81.78** | **69.05** |

## 6.4 Practical Applications of GDCLD

To evaluate the practical applicability of the CDCLD, we selected two significant landslide-triggering events that occurred in April 2024 for rapid landslide identification. These events include landslides induced by a heavy rainfall in Meizhou, China and landslides triggered by an earthquake in Hualien, China. In both cases, PlanetScope image was employed for

experimentation. For the Meizhou case, we obtained the image on May 14, 2024, and applied
SegFormer model trained on GDCLD data to identify landslides triggered by the heavy rainfall.
The results, shown in Figure.14, demonstrate that the GDCLD-trained model effectively
mapped newly-induced landslides with a total area of 8.49 $km^2$. The model exhibited excellent
accuracy in avoiding false positives such as buildings, roads, and rivers. In terms of the Hualien
event, we acquired post-event images from April 17 to 29, 2024. The rapid identification results,
displayed in Figure.15, indicate that the GDCLD-trained model effectively eliminates false
positives, such as roads, buildings, bare ground, and rivers, with the identified landslide area
of 90.9 $km^2$. The original PlanetScope images and landslide recognitions of the two events are
available at https://doi.org/10.5281/zenodo.13612636 (Fang et al., 2024)

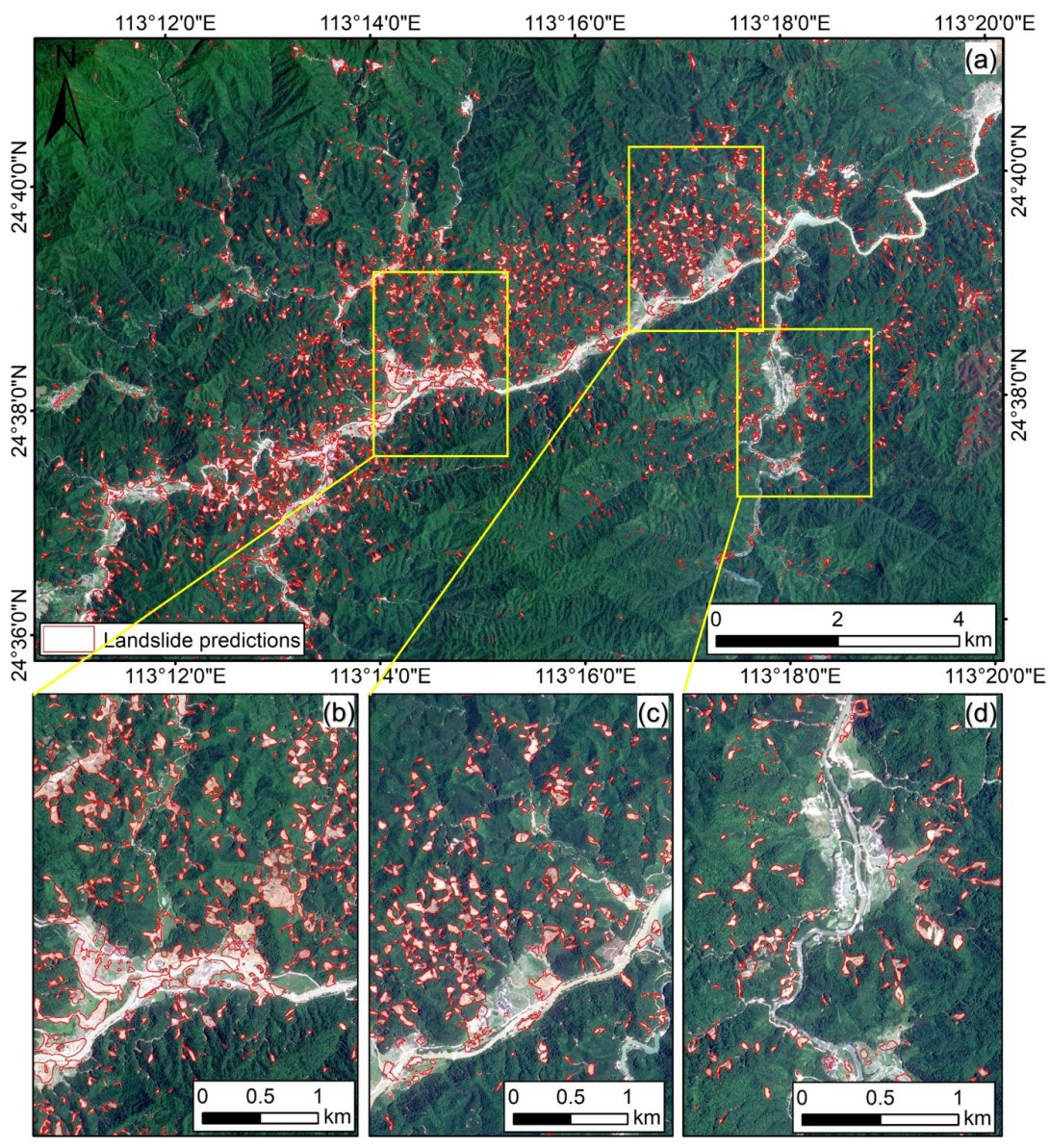


**Figure.14** Detection results of rainfall-induced landslides for Meizhou, China. (a) is the aerial

view of the whole area, (b), (c) and (d) is the partial details. Map credits: PlanetScope.


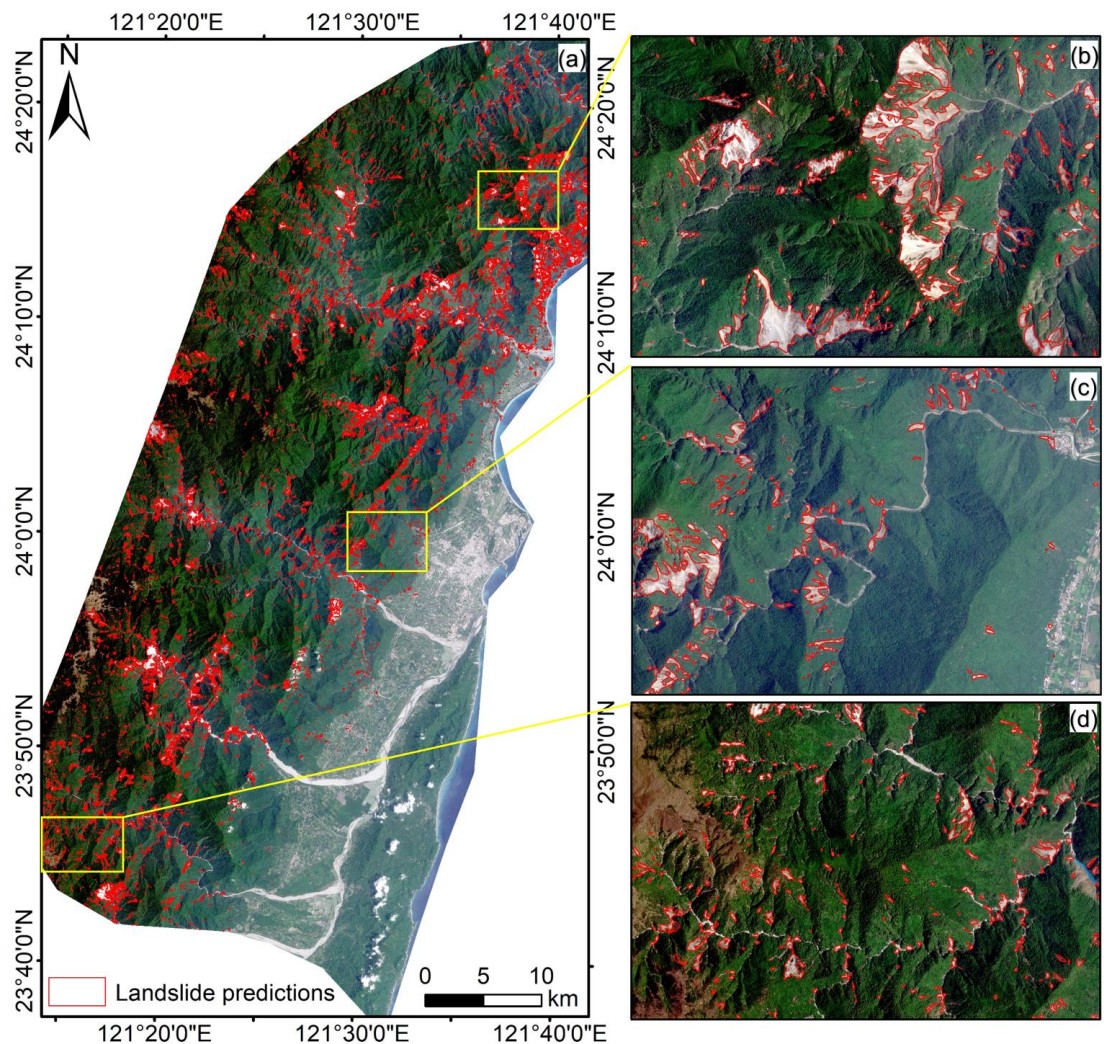

**Figure.15** Detection results of earthquake-triggered landslides for Hualien, China. (a) is the aerial view of the whole area, (b), (c) and (d) is the partial details. Map credits: PlanetScope.

# 7. Future research directions

The current GDCLD primarily comprises landslide samples from regions with significant vegetation coverage, with limited representation from areas with low vegetation cover, such as loess landslides. To address this, we have updated the database with high-resolution UAV data (0.1m resolution) of loess landslides triggered by the $M_w$ 6.2 earthquake in Jishishan, Gansu, China, in December 2023. Incorporating these loess landslide samples would enhance the dataset's diversity and improve the generalization capability of landslide detection models. Ongoing efforts to track and integrate data from landslides triggered by future extreme events

including strong earthquakes, heavy rainfall, and hurricanes, will further enrich the dataset.
In addition to expanding the GDCLD dataset, developing a large-scale vision model for
landslide detection, such as a Segment Anything Model tailored for landslide identification and
trained on GDCLD, is a crucial step forward in advancing AI-based landslide detection. This
model will be used for the intelligent recognition of landslides in multi-source remote sensing
image on a global scale.
Note that GDCLD is generally more applicable to semantic segmentation rather than
instance segmentation for landslide identification task. Unlike other instance segmentation
tasks, landslide segmentation presents unique challenges due to the frequent mixing of the
"deposit" areas of adjacent landslide bodies (Hungr et al., 2014). In most cases, we can only
intuitively identify the "source" area of a landslide. This phenomenon is commonly observed in
events such as the landslides triggered by the 2022 Luding earthquake in China (Figure.S10).
Under these circumstances, it is often not feasible to separate individual landslides directly from
2D optical images. Instead, it is necessary to consider the movement characteristics of each
object from a 3D perspective (Bhuyan et al., 2024; Marc and Hovius, 2015) and combine this
with topographic data to create accurate landslide labels for instance segmentation. However,
generating such datasets requires high-resolution digital elevation models (DEM) and UAV or
direct use of point cloud data. Given the global limitations in publicly available DEM (30m),
achieving such fine distinctions is challenging. Therefore, our current study primarily focuses
on semantic segmentation tasks. In future research, we plan to prepare landslide labels for
instance segmentation based on LiDAR observation, and to develop specialized algorithms to
address this complex issue.

# 702 8. Code and data availability

The data is freely available at https://doi.org/10.5281/zenodo.13612636 (Fang et al., 2024).
There are compressed folders, namely train_dataset.zip, val_dataset.zip and test_dataset.zip.
The train_dataset.zip file contains 11,162 TIFF-format RGB images and their corresponding
binary label data, with each image having dimensions of 1024×1024 pixels. The val_dataset.zip

file comprises 4,459 TIFF-format RGB images and binary label data, with each image also sized at 1024×1024 pixels. The test_data.zip file includes seven original remote sensing images from four landslide events, with images in TIFF-format RGB and labels in TIFF-format binary data, though the image dimensions vary. The Future work folder contains some remote sensing data that will be added later. For each label, "0" indicates the background, while "1" denotes the landslide. In addition, the other original data of UAV, Map World and Gaofen-6 are non-public data. Both the Map World and GF-6 datasets were accessed under an image license acquired by our team. The UAV data are under the usage rights of the laboratory affiliated with our team. If you need to use them, please contact the corresponding author. The original PlanetScope data were obtained through the Planet Education and Research Program. You can get original imageries at https://www.planet.com/ (Planet Team, 2019). And the code used to produce data described in this paper, as well as to create figures and tables, can be accessed at https://github.com/PaddlePaddle/PaddleSeg.

# 9. Conclusion

Landslide mapping across extensive geographic areas using remote sensing proves to be a significant challenge. Although previous attempts have produced landslide datasets and advanced automation and intelligence, they have not been able to overcome limitations of specific events and data sources. In this research, we proposed the Globally Distributed Coseismic Landslide Dataset (GDCLD), an innovative resource crafted to autonomously and precisely tackle the intricacies of landslide mapping. We made three significant contributions in this word. Firstly, we meticulously interpreted multi-source remote sensing data to create a comprehensive dataset for landslide detection. This dataset contains $1.39×10^9$ annotated landslide pixels and remote sensing image at four different resolutions, spanning nine global regions. It successfully addresses the crucial lack of large-scale datasets in current landslide identification research. Secondly, we utilized GDCLD -trained model to showcase its robustness and generalization in landslide identification across diverse geographical contexts. Our proposed dataset shows a great potential in rapid response and emergency management of

geological hazards. Although the landslide samples are obtained from seismic events, the trained model enable to capture and learn the characteristic differences between landslides and the surroundings, making them suitable for landslide mapping beyond seismic-triggered events, such as those caused by rainfall. The comparative analyses with existing datasets highlight its effectiveness as the data base of deep learning model in mapping landslides across various global regions. Finally, we demonstrate the superiority of the Transformer architecture over conventional CNN architecture in the task of landslide identification using multi-source remote sensing image. The GDCLD-S model further highlights the enhanced generalization capabilities of multi-source data compared to single-source data. This work has great practical implications for prevention and mitigation of geological hazard worldwide.

# Supplement

The supplement related to this article is available online at: XXXX

# Author contributions

All the authors contributed equally to the preparation of the paper, from data curation to the review of the final paper.

# Competing Interest

The authors declare that they have no known competing financial interests or personal relationships that could have appeared to influence the work reported in this paper.

# Disclaimer

Publisher's note: Copernicus Publications remains neutral with regard to jurisdictional claims in published maps and institutional affiliations.

# Acknowledgements

The research is supported by the National Science Fund for Distinguished Young Scholars of China (Grant No. 42125702), the National Natural Science Foundation of China (Grant No. 42307263), the New Cornerstone Science Foundation through the XPLORER PRIZE (Grant No. XPLORER-2022-1012), the Natural Science Foundation of Sichuan Province (Grant No. 2022NSFSC0003 and 2022NSFSC1083), and the China Scholarship Council (CSC NO. 202409230002). We would like to thank the State Key Laboratory of Geohazard Prevention and Geoenvironment Protection for providing UAV data, the National Platform for Common GeoSpatial Information Services for MAP WORLD data, and the China Centre for Resources Satellite Data and Application for Gaofen-6 data. We would like to thank Kushanav Bhuyan for helping improve the English and general writing in this paper. We sincerely thank all colleagues who contributed to the landslide interpretation work. Finally, we sincerely thank the Anonymous Reviewers for their precious time and insightful comments, which are very helpful for us to improve the quality and readability of our manuscript.

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
