# Peer review of "A globally distributed dataset of coseismic"

_Earth System Science Data, 2024_

## Author Comment (AC2)

*The Response to Comments from Review 1*

| Comment 1 |
| --- |
| Remove link and citation from the abstract of GDCLD dataset. |

| Response 1 |
| --- |
| Thanks for your careful comment. |
| We have removed the links and citations from the abstract. |

| Comment 2 |
| --- |
| Four independent regions are not clear. Authors must explain it clearly. |

| Response 2 |
| --- |
| Thank you for your constructive comment. In the first paragraph of **Section 5 (Results)** of the manuscript, we have modeified and provided a detailed introduction of the four independent regions. |
| *"To further assess the generalization ability of this dataset, we chose three types of remote sensing images (UAV, PlanetScope, and Map World image) from four independent seismic events (Lushan, Mestas, Sumatra, and Palu) as the test dataset. Considering the geographical distribution, these four regions, located on different continents and characterized by distinct tectonic settings and climatic conditions, ensure complete independence from the training dataset. From the perspective of data sources, the four study areas represent three major types of remote sensing imagery: PlanetScope, UAV, and Map World. Additionally, the UAV sensor used in the Lushan area is different from those used in other regions. This data partitioning strategy is designed to rigorously evaluate the generalization capability of the GDCLD-trained model." (P23L461~470)* |

| Comment 3 |
| --- |
| In the proposed work you have collected data for rainfall induced landslides or other parameters like topographical, anthropogenic and geological parameters are also considered? If yes mention it, if not what results will be observed after evaluating these parameters. |

| Response 3 |
| --- |
| Thanks for your valuable comment and suggestions. |
| 1. Firstly, the primary objective of this study is to develop a multi-source remote sensing dataset for the intelligent recognition of landslides in optical imagery. Consequently, other geological factors, including topography and geomorphology, were not considered in this research. While the incorporation of high-resolution Digital Elevation Model (DEM) data can indeed enhance the model's ability to recognize landslides, the publicly available DEM data has a resolution of 30 meters, which is too coarse when compared to high-resolution optical data. Therefore, this study has not yet considered incorporating topographic and geomorphological information extracted from DEMs. |
| 2. Furthermore, anthropogenic and geological parameters have not yet been incorporated into the dataset, despite their potential to enhance landslide detection accuracy. Regarding geological parameters, the currently available public data are too coarse to be directly applied to semantic segmentation tasks involving VHR remote sensing imagery. As for anthropogenic parameters, during the dataset creation process, we have included representations of human engineering activities as negative samples |

*(Figure.5)*, which can contribute to improving the generalization capability of landslide detection models.

3. Finally, regarding data on rainfall-induced landslides, this paper did not specifically collect such data. The experiments mentioned in the abstract on the intelligent recognition of rainfall-induced landslides mainly utilized remote sensing data from the GVLM dataset (Zhang et al., 2023) (*P34L619~624*), which includes rainfall-induced landslides. Additionally, during the revision process, we added a brief discussion in Section 6.4, where we incorporated PlanetScope data of rainfall-induced landslides that occurred in Meizhou and Guangzhou, China during April 2024. This data can be accessed at https://doi.org/10.5281/zenodo.11369484 (Fang et al., 2024).

**"*For the Meizhou case, we obtained the image on May 14, 2024, and applied SegFormer model trained on GDCLD data to identify landslides triggered by the heavy rainfall. The results are shown in Figure.14, demonstrating that the GDCLD-trained model can effectively map newly-induced landslides with a total area reached 8.49 km$^2$. The model shows excellent performance in avoiding false positives such as buildings, roads, and rivers.*" (P37L656~660)**

| Comment 4 |
| --- |
| Abstract written is so general, it must be rewritten highlighting the major objectives, method adopted and result achieved. |

**Response 4**

Thank you for raising this point. We have rewritten the abstract

*"Rapid and accurate mapping of landslides triggered by extreme events is essential for effective emergency response, hazard mitigation, and disaster management. However, the development of generalized machine learning models for landslide detection has been hindered by the absence of a high-resolution, globally distributed, event-based dataset. To address this gap, we introduce the Globally Distributed Coseismic Landslide Dataset (GDCLD), a comprehensive dataset that integrates multi-source remote sensing images, including PlanetScope, Gaofen-6, Map World, and Unmanned Aerial Vehicle data, with varying geographical and geological background for nine events across the globe. In this study, we evaluated the effectiveness of GDCLD by comparing the mapping performance of seven state-of-the-art semantic segmentation algorithms. These models were further tested by three different types of remote sensing images in four independent regions, while the GDCLD-SegFormer model get the best performance. Additionally, we extended the evaluation to a rainfall-induced landslide dataset, where the models demonstrated excellent performance as well, highlighting the dataset's applicability to landslide segmentation triggered by other factors. Our results confirm the superiority of GDCLD in remote sensing landslide detection modeling, offering a comprehensive data base for rapid landslide assessment following future unexpected events worldwide." (P2L16~32)*

| Comment 5 |
| --- |
| Too much old citations in the introduction section, it must be updated with latest citations like:
https://onlinelibrary.wiley.com/doi/abs/10.1002/ett.3998
https://link.springer.com/article/10.1007/s12145-022-00889-2 |

https://www.mdpi.com/2072-4292/16/6/992

https://www.nature.com/articles/s41597-023-02847-z

**Response 5**

Thanks for your careful review. We have updated the latest literature in the introduction section of the article.

**Comment 6**

Write paper organization at the end of the introduction section. Also write major objective of the paper achieved in the proposed work along with steps taken to accomplish the above objective.

**Response 6**

Thanks for your valuable advices.

We have added the corresponding content at the end of the Introduction.

*"The paper is structured as follows: Section 2 reviews existing high-quality landslide datasets to provide an overview of the current state of research. Section 3 introduces the data collection and preparation process to showcase the extensive research events and scientific methodology behind our data production. Section 4 describes the semantic segmentation algorithms, loss functions, and parameter settings used in this study, and shows their rationality. Section 5 presents the results, including the training, validation, and testing outcomes of the dataset, as well as the generalization ability of the GDCLD trained model in independent regions. Section 6 discusses the innovation and effectiveness of GDCLD, illustrating its effective application in several landslide events."* **(P5L96~104).**

**Comment 7**

In line 87 to 88 "Therefore, there is a pressing need for the development of a carefully curated and diverse dataset". It must be written properly.

**Response 7**

Thanks for your helpful comment. We have modified this section.

*"Therefore, there is an urgent need to develop a carefully curated and diverse dataset."* **(P4L87)**

**Comment 8**

Line 92 what kind of shortcomings were addressed. Have evaluated the existing dataset on the proposed method. If yes then kindly share the result. If not, evaluate it, and add one table highlighting the same.

**Response 8**

Thanks for your helpful comments.

Our study addresses the following key shortcoming:

1. The GDCLD dataset, constructed from multi-source remote sensing imagery, represents the largest publicly available landslide dataset to date. Unlike existing datasets, GDCLD is entirely composed of high-resolution remote sensing imagery, with annotated pixels reaching a total of 1.39 billion. Furthermore, it incorporates negative samples such as bare land, cloud cover, dry riverbeds, and human engineering activities. These features significantly enhance the generalization capability of models trained on GDCLD. This aspect is elaborated in **Section 3.2** and is further substantiated by the model's excellent generalization performance as demonstrated in **Sections 5 and 6.3**.

Additionally, compared to other publicly available but smaller landslide datasets, GDCLD effectively supports the training of large neural networks based on the Transformer architecture, providing a robust data foundation for the future development of large vision models for landslide detection. This aspect is further discussed in *Section 7*.

2. To demonstrate the performance of GDCLD, we have supplemented the study with relevant experiments, providing a detailed comparison of existing landslide datasets in *Section 6.3*. As presented in *Table 10*, the models trained on GDCLD exhibit superior performance compared to those trained on other datasets, underscoring the advantages of GDCLD.

**Comment 9**
Section 2 must be written as "Related Work"

**Response 9**
Thanks for your comment. We have modified *"Relate Work"* to *"Related Work"*. *(P5L105)*

**Comment 10**
In section 3 except "Data Collection" all other subsections must be presented in tabular form rather in running text.

**Response 10**
Thank you for your positive and constructive comments.

In *Section 3.1*, following your suggestion, we have organized the nine earthquake-induced landslide events collected in this study into a table. The table provides a clear overview of each event, displaying key information such as the date, magnitude, geographic location, number of landslides, and total landslide area. This approach allows for an intuitive comparison of all events, enabling readers to easily grasp the critical details.

Table.2 Summary table of landslide event information in GDCLD

| Events | Mw | time | Geographic coordinates | Landslide number | Total landslide area (km$^2$) |
|---|---|---|---|---|---|
| Jiuzhaigou | 6.5 | 2017 | (102.82°E, 33.20°N) | 2498 | 14.5 |
| Mainling | 6.4 | 2017 | (95.02°E, 29.75°N) | 1448 | 33.6 |
| Hokkaido | 6.6 | 2018 | (142.01°E, 42.69°N) | 7962 | 23.8 |
| Palu | 7.5 | 2018 | (119.84°E, 0.18°S) | 15700 | 43.0 |
| Mesetas | 6.0 | 2019 | (76.19°W, 3.45°N) | 804 | 8.5 |
| Nippes | 7.2 | 2021 | (73.45°W, 18.35°N) | 4893 | 45.6 |
| Sumatra | 6.1 | 2022 | (100.10°E, 0.22°N) | 602 | 10.6 |
| Lushan | 5.9 | 2022 | (102.94°E, 30.37°N) | 1063 | 7.2 |
| Luding | 6.8 | 2022 | (102.08°E, 29.59°N) | 15163 | 28.53 |

*(P9L179)*

We also retain the detailed descriptions of each event in the main text. These running texts provide a more comprehensive understanding of the events, including background information, geographic and geological contexts, and specifics regarding the multi-source remote sensing imagery. This detailed narrative enriches the reader's perspective and enhances their understanding of the study's context.

**Comment 11**

Section 3.2 highlights the preprocessing of the dataset. One detailed fig must be added highlighting the steps involved or operations performed on training dataset.

**Response 11**

Thanks for your advice.

We have drawn a flowchart of the dataset preprocessing and added it ***Section 3 (Figure.1)***. *(P8L164~171)*

*"The creation of the GDCLD dataset can be broadly divided into two main components: landslide data collection and remote sensing data processing. In the first part, we compiled recent landslide events triggerred by earthquakes worldwide over the past seven years and obtained the corresponding remote sensing image. The second part details the process of annotating landslide labels and the methodology used to create the standard dataset. The workflow is illustrated in Figure.1.*

[Figure]

*Figure.1 The workflow of producing GDCLD"*

**Comment 12**

Mention the technical novelty of the paper other than creating the generalized dataset.

**Response 12**

Thank you for the suggestion.

This study primarily focuses on innovation in dataset development, aiming to provide a high-quality landslide dataset for research on intelligent landslide recognition, contributing to disaster prevention and mitigation efforts alongside researchers worldwide. Therefore, we have integrated state-of-the-art remote sensing and computational techniques that are currently open-source, with a particular emphasis on the contributions and innovations of the GDCLD dataset.

1. GDCLD is a landslide dataset based on multi-source, multi-sensor, and cross-resolution high-precision remote sensing imagery, entirely annotated manually. It is suitable for intelligent landslide recognition tasks across a wide range of scenarios.

2. The dataset encompasses multiple global events, spanning various climate zones,

tectonic settings, and geomorphological landscapes, and features an extended temporal resolution.

3. We have manually incorporated a rich set of negative samples, such as bare land, exposed rock, dry riverbeds, cloud cover, and human engineering activities, which are prone to confusion with landslides. This work was carried out to enhance the generalization capability of models trained on this dataset.

4. During the manual annotation process, we meticulously cross-referenced the spectral characteristics of pre- and post-event remote sensing imagery with the morphological features of landslides. Additionally, we conducted field surveys for certain landslide events.

5. In terms of intelligent recognition, we conducted the first comparative analysis of different neural network architectures on a multi-source remote sensing landslide dataset, confirming the superiority of the Transformer architecture for landslide recognition.

6. Regarding remote sensing data sources, we compared the effectiveness of single-source versus multi-source remote sensing imagery in landslide recognition, demonstrating the effectiveness of multi-source imagery in such tasks.

We mentioned the technical innovations of this study in the conclusion:

*"Finally, we demonstrate the superiority of the Transformer architecture over conventional CNN architecture in the task of landslide identification using multi-source remote sensing image. The GDCLD-S model further highlights the enhanced generalization capabilities of multi-source data compared to single-source data."* **(P42L740~743)**
* * *
**Comment 13**

How can the GDCLD and the trained models be integrated into current emergency response and disaster management systems? Are there any case studies or real-world applications that demonstrate their effectiveness?
* * *
**Response 13**

Thanks for your valuable questions.

Regarding the geological disaster emergency identification system, we integrated this identification model into our institute, the laboratory's emergency system (SKLGP-LDD). In order to show the applications of GDCLD in landslides triggered by real events, we added a discussion **section 6.4 "Practical Applications of GDCLD"** and conducted rapid mapping of two landslide events that occurred in 2024.

*"To evaluate the practical applicability of the CDCLD, we selected two significant landslide-triggering events that occurred in April 2024 for rapid landslide identification. These events include landslides induced by a heavy rainfall in Meizhou, China and landslides triggered by an earthquake in Hualien, China. In both cases, PlanetScope image was employed for experimentation. For the Meizhou case, we obtained the image on May 14, 2024, and applied SegFormer model trained on GDCLD data to identify landslides triggered by the heavy rainfall. The results, shown in Figure 14, demonstrate that the GDCLD-trained model effectively mapped newly-induced landslides with a total area of 8.49 km2. The model exhibited excellent accuracy in avoiding false positives such as buildings, roads, and rivers. In terms of the Hualien event, we acquired post-*

*event images from April 17 to 29, 2024. The rapid identification results, displayed in Figure.15, indicate that the GDCLD-trained model effectively eliminates false positives, such as roads, buildings, bare ground, and rivers, with the identified landslide area of 90.9 km2. The original PlanetScope images and landslide recognitions of the two events are available at https://doi.org/10.5281/zenodo.13612636 (Fang et al., 2024)"* **(P36~39L652~671)**

[Figure]

*Figure.14 Detection results of rainfall landslides for Meizhou, China. (a) is the aerial view of the whole area, (b), (c) and (d) is the partial details.*

[Figure]

***Figure.15*** *Detection results of earthquake-triggered landslides for Hualien, China. (a) is the aerial view of the whole area, (b), (c) and (d) is the partial details.*

**Comment 14**

What are the challenges and considerations for scaling this approach to cover larger areas or more diverse regions? Are there any technological or infrastructural requirements?

**Response 14**

Thanks for your valuable comment.

1. At first, the GDCLD dataset is currently most effective in research areas with moderate vegetation cover, as it successfully mitigates interference based on negative samples such as clouds, bare land, and dried riverbeds. However, its application in detecting loess landslides, such as those triggered by the Mw 6.2 earthquake in Jishishan, Gansu, China, on December 18, 2023, exhibits certain limitations in our approach. As we mentioned in section 7, we will continue to expand our dataset in the future to enable it to meet the needs of a wider range of landslide identification tasks.

*"The current GDCLD primarily comprises landslide samples from regions with significant vegetation coverage, with limited representation from areas with low vegetation cover, such as loess landslides. To address this, we have updated the database with high-resolution UAV data (0.1m resolution) of loess landslides triggered by the $M_w$ 6.2 earthquake in Jishishan, Gansu, China, in December 2023. Incorporating these loess landslide samples would enhance the dataset's diversity and improve the generalization capability of landslide detection models. Ongoing efforts to track and integrate data from landslides triggered by future extreme events including strong earthquakes, heavy rainfall, and hurricanes, will further enrich the dataset."*

*(P39~40L673~680)*

2. In addition, to meet the needs of automatic landslide identification in a larger area, a larger neural network model is needed. This requires not only accurate training data from geological researchers, but also sufficient computing power and computer science. We also mentioned in **section 7** that we will train a large visual model based on GDCLD in the future.

*"In addition to expanding the GDCLD dataset, developing a large-scale vision model for landslide detection, such as a Segment Anything Model tailored for landslide identification and trained on GDCLD, is a crucial step forward in advancing AI-based landslide detection. This model will be used for the intelligent recognition of landslides in multi-source remote sensing image on a global scale" **(P40L681~685)***

**Comment 15**

What are the potential future enhancements or expansions planned for the GDCLD? Are there any ongoing efforts to continuously update and improve the dataset?

**Response 15**

Thanks for your questions.

In **Section.7**, we outline several future research directions, including the expansion of the dataset. We plan to track landslide events triggered by future extreme events and incorporate them into our multi-source landslide dataset. During the current revision stage, we have already added PlanetScope data for two landslide events in Hualien and Meizhou, China. Notably, for dataset expansion, we have also included a UAV-based (0.1m resolution) dataset of earthquake-induced landslides in the Loess region, which will significantly enhance the richness and diversity of the GDCLD dataset. The dataset and detailed data description can be download from **https://doi.org/10.5281/zenodo.13612636 (Fang et al., 2024).**

**Comment 16**

What specific characteristics of the GDCLD-trained model enable it to effectively map rainfall-induced landslides? Are there any limitations or areas for improvement in this application?

**Response 16**

We thank the reviewer for raising these points.

1. Models trained on the GDCLD dataset are capable of distinguishing landslides from surrounding objects by learning differences in spatial morphology, spectral characteristics, etc. Given that both earthquake- and rainfall-induced landslides are typically newly induced landslides, they often exhibit significant spectral and spatial contrasts with their surrounding environment, making them feasible to be identified.

2. However, since some rainfall-induced landslides are shallow and do not fully disrupt the vegetation cover, the model's performance in detecting this type of landslides may be suboptimal (Wang et al., 2022). In future work, we plan to specifically address this limitation by augmenting the dataset with more samples of rainfall-induced landslides, aiming at improving the generalization capability of the GDCLD model.

**Comment 17**

How does the performance of the GDCLD-trained model compare to existing models and datasets in quantitative terms? Can you include specific performance metrics or visual comparisons?

**Response 17**

Thanks for your valuable comments.

In order to further demonstrate the advantages of GDCLD over other landslide datasets, we modified *section 6.3* by adding the GVLM- and CAS-trained models based on the SegFormer algorithm. Futhermore, we also reproduced the CAS-D model trained with DeepLabV3 in the CAS data paper *(*Xu et al., 2024*)*. These models were implemented for landslide identification task with three different remote sensing data sources in the independent 2022 Lushan case study in this paper. In Section 6.3, we have supplemented the discussion with relevant experimental data:

1. As observed in *Table 10*, the CAS-D model demonstrates mIoU results of 57.91% on UAV image, 52.86% on PlanetScope image, and 58.11% on Map World image within the Lushan dataset. Overall, these results are inferior to the performance of GDCLD-S on the Lushan dataset. Additionally, CAS-D's performance lags behind that of CAS-S, which is based on the Transformer architecture.

2. *Table 10* also highlights the performance of landslide detection models trained with the SegFormer architecture on the GVLM, CAS, and GDCLD datasets. Among them, GDCLD-S exhibits the highest performance, with mIoU results of 72.96% for UAV, 69.05% for PlanetScope, and 71.92% for MapWorld image, underscoring the superior competitiveness of the GDCLD dataset.

The overall changes to *section 6.3* are as follows: *(P33~36L602~650)*

*"6.3 Comparison with existing landslide datasets and models*

[revised manuscript text omitted]

*"*

**Comment 18**

Which seven semantic segmentation algorithms were evaluated, and what were the criteria for their selection? How do these algorithms differ in their approach to landslide detection?

**Response 18**

Thanks for your comments.

1. In this study, we selected seven semantic segmentation algorithms—UNet, ResUNet, DeepLabV3, HRNet, UperNet, SwinUNet, and SegFormer (Tang et al., 2022; Meena et

al., 2022; He et al., 2022; Li et al., 2022) *(Section 4.1)*. The first four algorithms are based on a pure CNN architecture, while the latter three are based on a Transformer architecture. These algorithms have been among the most popular for semantic segmentation tasks during different periods, which have been applied to various remote sensing tasks, including landslide detection. Typically, CNN-based algorithms are well-suited for small datasets, whereas Transformer-based algorithms perform better on larger datasets. Therefore, we chose these seven semantic segmentation algorithms to comprehensively evaluate the GDCLD dataset.

*2*. Regarding the performance of the seven semantic segmentation algorithms in landslide detection, experimental results from the GDCLD validation and test sets indicate that semantic segmentation models based on the Transformer architecture outperform those based purely on CNN architectures in multi-source remote sensing image recognition tasks. The results are presented in ***Tables.4 and 5***. This superiority can be attributed to the Transformer models' larger receptive fields, which enable the effective learning of high-level features from multi-source imagery, thereby enhancing their generalization capabilities. ***In section 5***, we give a more detailed explanation. ***(P23~26L458~543)***

**Table.4** Comparison of result on GDCLD validation dataset

| Method | Backbone | Precision (%) | Recall (%) | F1 (%) | mIoU (%) |
|---|---|---|---|---|---|
| UNet | - | 77.05 | 82.01 | 79.54 | 71.07 |
| ResUNet | ResNet-50 | 78.17 | 86.48 | 82.11 | 71.94 |
| DeepLabV3 | ResNet-50 | 81.27 | 86.96 | 84.02 | 74.61 |
| HRNet | HRNet-48 | 81.88 | 87.21 | 84.46 | 75.19 |
| UperNet | ViT-B16 | 88.18 | 90.64 | 89.39 | 81.97 |
| SwinUNet | - | 89.78 | **92.01** | 90.72 | 83.68 |
| SegFormer | MiT-B4 | **91.35** | 91.70 | **91.52** | **85.06** |

**Table.5** Comparison of result on test dataset

| Method | Backbone | Precision (%) | Recall (%) | F1 (%) | mIoU (%) |
|---|---|---|---|---|---|
| UNet | - | 61.69 | 61.22 | 61.45 | 56.09 |
| ResUNet | ResNet-50 | 66.56 | 64.46 | 65.49 | 57.06 |
| DeepLabV3 | ResNet-50 | 65.26 | 67.75 | 66.48 | 59.73 |
| HRNet | HRNet-48 | 65.52 | 72.03 | 68.62 | 61.79 |
| UperNet | ViT-B16 | 69.96 | 78.08 | 73.80 | 65.42 |
| SwinUNet | - | 71.56 | 82.26 | 76.54 | 67.18 |
| SegFormer | MiT-B4 | **77.09** | **87.09** | **81.88** | **72.84** |

**Comment 19**

Why were PlanetScope, Gaofen-Map World, and Unmanned Aerial Vehicles chosen as the primary sources of remote sensing images? Are there other potential sources that could be included in future iterations of the dataset?

**Response 19**

We thank the reviewer for raising the questions.

1. At first, considering that this study utilizes very high-resolution (VHR) remote sensing imagery, with a spatial resolution of 3m for PlanetScope, 2m for Gaofen-6, 0.5-1m for Map World, and 0.2m for UAV, the overall resolution range of the multisource remote sensing images spans from 0.2m to 3m. This range effectively encompasses the entire spectrum of VHR spatial resolutions, enabling GDCLD to meet the demands of landslide detection tasks in most scenarios.

2. In future work, we plan to further enrich GDCLD with additional remote sensing data sources, including Gaofen-2 (0.8m), Gaofen-7 (0.5m), Google Earth image (0.5m-1m), and higher-resolution UAV (0.1~0.2m), to support the development of even more accurate landslide detection models. We have updated some of the UAV data of the landslides induced by the Loess Earthquake with a spatial resolution of 0.1m, which can be downloaded from **https://doi.org/10.5281/zenodo.13612636 (Fang et al., 2024)**.

"*To address this, we have updated the database with high-resolution UAV data (0.1m resolution) of loess landslides triggered by the Mw 6.2 earthquake in Jishishan, Gansu, China, in December 2023.*" ***(P39L675~677)***

**Comment 20**

What specific criteria and methods were used to annotate the 1.39 billion landslide pixels? Were there any challenges or limitations encountered during the annotation process?

**Response 20**

Thanks for your comments.

1. During the annotation of the 1.39 billion landslide pixels, as illustrated in ***Figure.1 and Figure.2***, we first considered the spectral changes in remote sensing images before and after the event to preliminarily identify landslide-affected areas. Subsequently, we incorporated topography, landforms, and surrounding objects to assess the morphological characteristics of the landslides, allowing us to exclude images of bare land, vegetation changes over time, river changes, and human engineering activities (Fan et al., 2019). Using QGIS software, vector labels were meticulously drawn for each landslide pixel.

2. The annotation process encountered significant challenges, including interference from exposed bedrock and bare land, as well as debris accumulation at channel outlets caused by debris flows. To address the first challenge, we meticulously compared the spectral and morphological characteristics of pre- and post-earthquake remote sensing images to eliminate the interference from non-landslide features, such as pre-existing bare land. For the second challenge, we focused on the morphological differences between landslides and debris flows to mitigate the impact of debris flow channels and deposition areas on spectral changes in remote sensing image (Hungr et al., 2014).

---

## Author Comment (AC3)

**The Response to Comments from Review 2**

*Comment 1*

*Table 1, please provide the number of sites where landslides have occurred, along with the number of landslide polygons for each dataset.*

*Response 1*

*Thanks for your careful suggestion.*

*We have revised Table 1 to include the specific events corresponding to each dataset and the number of landslides associated with each event. However, due to the absence of detailed information in the original sources for the GVLM and CAS datasets, some data remain unavailable, resulting in incomplete information.* **(P7L161)**

*Table.1 Existing landslide dataset statistics*

| Dataset | Bands | events | Tiles | Landslides number | Labeling pixels |
|---|---|---|---|---|---|
| Bijie landslide | 3 | 1 | 2773 | 770 | $7.23 \times 10^6$ |
| Landslide4sense | 14 | 4 | 3799 | >30000 | $1.76 \times 10^6$ |
| HR-GLDD | 4 | 13 | 1756 | 7193 | $2.96 \times 10^6$ |
| GVLM | 3 | 17 | 17 | - | $3.24 \times 10^7$ |
| CAS Landslide | 3 | 9 | 20865 | - | $1.95 \times 10^8$ |

*Comment 2*

*Table 2, please specify the total number of polygons obtained and confirms that the necessary rights for the use of the mentioned images.*

*Response 2*

*Thanks for your valuable comment.*

*1. We have made some revisions to the content of Section 3.1, "Data Collection." In response to your suggestions, we have added Table 2 to provide additional information. This table includes details such as the time, geographic coordinates (latitude and longitude), the number of landslides, and the total area affected by each landslide event.* **(P9L179)**

*Table.2 Summary table of landslide event information in GDCLD*

| Events | Mw | time | Geographic coordinates | Landslide number | Total landslide area (km$^2$) |
|---|---|---|---|---|---|
| Jiuzhaigou | 6.5 | 2017 | (102.82°E, 33.20°N) | 2498 | 14.5 |
| Mainling | 6.4 | 2017 | (95.02°E, 29.75°N) | 1448 | 33.6 |
| Hokkaido | 6.6 | 2018 | (142.01°E, 42.69°N) | 7962 | 23.8 |
| Palu | 7.5 | 2018 | (119.84°E, 0.18°S) | 15700 | 43.0 |
| Mesetas | 6.0 | 2019 | (76.19°W, 3.45°N) | 804 | 8.5 |
| Nippes | 7.2 | 2021 | (73.45°W, 18.35°N) | 4893 | 45.6 |
| Sumatra | 6.1 | 2022 | (100.10°E, 0.22°N) | 602 | 10.6 |
| Lushan | 5.9 | 2022 | (102.94°E, 30.37°N) | 1063 | 7.2 |
| Luding | 6.8 | 2022 | (102.08°E, 29.59°N) | 15163 | 28.53 |

2. Details regarding data authorization are provided in Section 8, "Data Availability." The Planet data were obtained through the Planet Education and Research Program. Both the Map World and GF-6 datasets were accessed under an image license acquired by our team. The UAV data are under the usage rights of the laboratory affiliated with our team.

"Both the Map World and GF-6 datasets were accessed under an image license acquired by our team. The UAV data are under the usage rights of the laboratory affiliated with our team. If you need to use them, please contact the corresponding author. The original PlanetScope data were obtained through the Planet Education and Research Program. You can get original imageries at https://www.planet.com/ (Planet Team, 2019)." (P41L713~717)

**Comment 3**

In Fig. 4, it's crucial to clarify the distinction between 'Label' and 'Ground Truth,' as they may initially appear similar.

**Response 3**

Thank you for giving this comment.

In Figure.4, the "label" represents binary pixel value derived from manually interpreted landslide polygons, while the "ground truth" is depicted by overlaying the semi-transparent landslide label on the corresponding position of the image. This approach visually

*demonstrates the accuracy of our landslide annotations. In the caption of Figure.4, we added a sentence to explain these words.*

*"The "label" refers to the binary landslide mask, whereas the "Ground Truth" illustrates the concordance between the annotated and actual landslide in images." (P19L351~353)*

**Comment 4**

*A clear workflow outlining the entire dataset production process, along with details on personnel involvement, costs, and time invested, would offer valuable insights into the significant effort required to create such a comprehensive resource.*

**Response 4**

*Thanks for your insightful advices.*

*1. We have drawn a flowchart of the dataset preprocessing and added it **Section 3 (Figure.1)**. **(P8L164~171)***

"The creation of the GDCLD dataset can be broadly divided into two main components: landslide data collection and remote sensing data processing. In the first part, we compiled recent landslide events induced by earthquakes worldwide over the past seven years and obtained the corresponding remote sensing imagery. The second part details the process of annotating landslide labels and the methodology used to create the standard dataset. The workflow is illustrated in Figure.1.

[Figure]

Figure.1 The workflow of producing GDCLD"

*2. Regarding the specific timeline and procedures for dataset creation, the landslide data included in the GDCLD were interpreted by our team over one year of research.*

"The mapping of landslide polygons for these nine events was primarily conducted by a team of five researchers, including the authors. All team members possess expertise in geology or remote sensing and were involved in a year-long process of detailed interpretation." **(P13~14L298~301)**

*Moreover, we have acknowledged the efforts of all colleagues involved in the landslide interpretation in the* **Acknowledgements** *section with the following statement: "We sincerely thank all colleagues who contributed to the landslide interpretation work."* **(P43L768~769)**
* * *
**Comment 5**

*Lastly, the section titled '6.3 Model based on GDCLD performance on existing datasets' necessitates clarification to ensure its content is fully understood.*
* * *
**Response 5**

*We thank the reviewer for raising the question.*

*During the revision of our manuscript, we have made adjustment to the content of* **Section 6.3** *and also revised its title. The overall content of* **Section 6.3** *is outlined as follows:* **(P33~36L601~650)**

"6.3 Comparison with existing landslide datasets and models

[revised manuscript text omitted]

---

## Author Comment (AC4)

**The Response to Comments from Review 3**

**Comment 1**

However, a significant concern with this work, as with any automated landslide mapping, is the potential clustering of multiple landslides in one location, leading to the incorrect identification of several landslides as a single event. For instance, in the Hokkaido landslide area, several crowns have merged, resulting in a unified depositional landscape. Was any attempt made to address this issue by separating the multiple landslides? If not, this should be discussed in the limitations section and considered for future work.

**Response 1**

Thank you for giving this comment. We strongly agree with your opinion that the current available landslide datasets are all facing such challenge. Currently, most publicly available landslide datasets are designed for semantic segmentation tasks rather than instance segmentation. Unlike other computer vision tasks, landslides are complex geological phenomena, and distinguishing multiple landslides that overlap or blend together is challenging when relying solely on optical imagery. Effective separation often requires additional data, such as digital elevation models (DEMs) and derived geomorphological features. We have addressed this issue in Section 7 and plan to develop a dedicated landslide instance segmentation dataset in future work.

"*Note that GDCLD is generally more applicable to semantic segmentation rather than instance segmentation for landslide identification task. Unlike other instance segmentation tasks, landslide segmentation presents unique challenges due to the frequent mixing of the "deposit" areas of adjacent landslide bodies (Hungr et al., 2014). In most cases, we can only intuitively identify the "source" area of a landslide. This phenomenon is commonly observed in events such as the landslides triggered by the 2022 Luding earthquake in China (Figure.S10). Under these circumstances, it is often not feasible to separate individual landslides directly from 2D optical images. Instead, it is necessary to consider the movement characteristics of each object from a 3D perspective (Bhuyan et al., 2024; Marc and Hovius, 2015) and combine this with topographic data to create accurate landslide labels for instance segmentation. However, generating such datasets requires high-resolution digital elevation models (DEM) and UAV or direct use of point cloud data. Given the global limitations in publicly available DEM (30m), achieving such fine distinctions is challenging. Therefore, our current study primarily focuses on semantic segmentation tasks. In future research, we plan to prepare landslide labels for instance segmentation based on LiDAR observation, and to develop specialized algorithms to address this complex issue. (P40L686~701)*

[Figure]

Merging label   Dividing label   Landslide 1

Landslide 2   Landslide 3   Landslide 4

*Figure.S10 Example of instance landslide label (2022 Luding earthquake-triggered landslides)*
"

| Comment 2 and Comment 3 |
| --- |
| 1. In the abstract, specify the number of events or case areas represented by this global ML-based inventory. |
| 2. On line 27, mention the best-fit model used. |

**Response 2 and Response 3**

Thanks for your insightful advices.

We have completely rewritten the summary and added content based on your suggestions.

*"Rapid and accurate mapping of landslides triggered by extreme events is essential for*

*effective emergency response, hazard mitigation, and disaster management. However, the development of generalized machine learning models for landslide detection has been hindered by the absence of a high-resolution, globally distributed, event-based dataset. To address this gap, we introduce the Globally Distributed Coseismic Landslide Dataset (GDCLD), a comprehensive dataset that integrates multi-source remote sensing images, including PlanetScope, Gaofen-6, Map World, and Unmanned Aerial Vehicle data, with varying geographical and geological background for nine events across the globe. In this study, we evaluated the effectiveness of GDCLD by comparing the mapping performance of seven state-of-the-art semantic segmentation algorithms. These models were further tested by three different types of remote sensing images in four independent regions, while the GDCLD-SegFormer model get the best performance. Additionally, we extended the evaluation to a rainfall-induced landslide dataset, where the models demonstrated excellent performance as well, highlighting the dataset's applicability to landslide segmentation triggered by other factors. Our results confirm the superiority of GDCLD in remote sensing landslide detection modeling, offering a comprehensive data base for rapid landslide assessment following future unexpected events worldwide.*" (P2L 16~32)

**Comment 4**

On line 69, where it is stated that most models lack generalization capability across diverse environmental backgrounds and remote sensing images, please elaborate on what the authors mean by "generalization."

**Response 4**

Thank you for giving this comment.

In this point, the term "generalization ability" refers to the capacity of machine learning or deep learning algorithms to adapt to new and unseen samples. This aims to illustrate that most models trained on existing datasets experience a significant decline in landslide detection performance when confronted with different geographic regions and remote sensing data sources. In the revised Section 6.3, we have included experiments to substantiate this observation. Specifically, we evaluated the ability of models trained on three datasets—CAS, GVLM, and GDCLD—to detect landslides in previously unseen areas.

"*In addition to the aforementioned analyses, we compare the performance of GDCLD with other two datasets, GVLM and CAS. Specifically, we train landslide detection models using the SegFormer algorithm on the GVLM and CAS datasets, denoted as GVLM-S and CAS-S, respectively, with identical training parameters as previously described. Furthermore, we also use the DeepLabV3 to train the CAS-D model based on the CAS dataset and use it for comparison of landslide detection (Xu et al., 2024). Subsequently, the GDCLD-S, CAS-S, CAS-D and GVLM-S models were applied to identify landslides in the Lushan area using three distinct remote sensing data sources: UAV, PlanetScope, and Map World. The results of this comparison are presented in Table 10. From Table 10, it is evident that the GDCLD-S model outperformed CAS-S, CAS-D and GVLM-S across all three remote sensing datasets, achieving mIoU of 72.96%, 69.05%, and 71.92% on UAV, PlanetScope, and Map World. In contrast, CAS-S records mIoU values of 62.03%, 56.86%, and 60.35% for the same*

*datasets, respectively, which is better than the CAS-D model trained with DeepLabV3, and also illustrates the advantages of the transformer architecture over the CNN architecture. Notably, GDCLD-S exhibited a significantly higher Recall than the other two models and also demonstrated an advantage in Precision. Overall, GDCLD-S, along with CAS-S, exhibited superior performance compared to the single-source data model GVLM-S, particularly in handling multisource remote sensing images. The extensive landslide data and negative samples included in GDCLD-S further contributed to its enhanced robustness against noise and improved Recall in landslide detection.* **(P35~36L629~648)**

**Table.10** Performance of the GDCLD-S, GVLM-S, CAS-S, and CAS-D models on the 2022 Lushan case

| Model | Data type | Precision (%) | Recall (%) | F1 (%) | mIoU (%) |
|---|---|---|---|---|---|
| CAS-D | UAV | 72.73 | 55.34 | 62.88 | 57.91 |
| | PlanetScope | 52.07 | 56.05 | 53.93 | 52.86 |
| | Map World | 61.79 | 70.50 | 64.9 | 58.11 |
| GVLM-S | UAV | 73.03 | 54.84 | 57.67 | 53.41 |
| | PlanetScope | 60.13 | 53.40 | 54.82 | 51.52 |
| | Map World | **77.71** | 66.40 | 71.56 | 63.97 |
| CAS-S | UAV | 74.08 | 67.05 | 69.95 | 62.03 |
| | PlanetScope | 58.56 | 76.57 | 66.40 | 56.86 |
| | Map World | 75.02 | 64.65 | 68.37 | 60.35 |
| GDCLD-S | UAV | **74.72** | **90.35** | **81.80** | **72.96** |
| | PlanetScope | **81.50** | **82.28** | **81.78** | **69.05** |
| | Map World | 76.18 | **87.35** | **81.38** | **71.92** |

"

**Comment 5**

On line 74, consider starting the sentence with "For instance" or "For example.""

**Response 5**

Thanks for your careful comment.

We have modified this word. "*for example, after major events such as the Wenchuan, China (2008), and Gorkha, Nepal (2015) earthquakes.*" **(P4L62~64)**

**Comment 6**

It is advisable to use the full forms of abbreviations like CAS, HRGLDD, and GVLM at their first occurrence in the manuscript.

**Response 6**

Thanks for your careful comment.

| We have reviewed the article and revised the content. |
|---|

**Comment 7**

A flowchart detailing the method would be helpful for readers.

**Response 7**

Thanks for your insightful advices, which will improve our work a lot.

We have drawn a flowchart of the dataset preprocessing and added it *Section 3 (Figure.1)*. *(P8L164~171)*

*"The creation of the GDCLD dataset can be broadly divided into two main components: landslide data collection and remote sensing data processing. In the first part, we compiled recent landslide events triggerred by earthquakes worldwide over the past seven years and obtained the corresponding remote sensing imagery. The second part details the process of annotating landslide labels and the methodology used to create the standard dataset. The workflow is illustrated in Figure.1.*

[Figure]

*Figure.1 The workflow of producing GDCLD"*

**Comment 8**

On line 96, consider changing the heading to "Related Work" or "Past Research.".

**Response 8**

Thanks for your comment. We have modified *"Relate Work"* to *"Related Work"*. *(P5L105)*

**Comment 9**

On line 97, the intended meaning is unclear and needs clarification.

**Response 9**

Thanks for your careful comment. We have modified *this word*.

*"The most effective approach for landslide mapping currently involves image segmentation, and computer vision segmentation tasks depend heavily on high-quality data to build accurate models. However, landslide segmentation tasks have developed relatively recently compared to other computer vision applications, resulting in only a limited number of studies that have constructed datasets for various landslide events. In*

*this section, we review some of these landslide segmentation datasets and provide detailed information on each (Table.1)."* **(P5L106~111)**

**Comment 10**

On line 198, change the reference to "Hokkaido earthquake."

**Response 10**

Thanks for your careful comment. We have modified *this word*.

"*Following the Hokkaido earthquake, we acquired PlanetScope image with a 3m resolution on December 12, 2018, and Map World image with a 0.5m resolution (Figure.S3)."* **(P10~11L215~217)**

**Comment 11**

The source for the World Map image, as well as other data sources, such as the download link or web portal, should be mentioned for the readers.

**Response 11**

We thank the reviewer for raising these points.

In section 8, we modified the content of Data availability and introduced the source of the dataset.

"*In addition, the other original data of UAV, Map World and Gaofen-6 are non-public data. Both the Map World and GF-6 datasets were accessed under an image license acquired by our team. The UAV data are under the usage rights of the laboratory affiliated with our team. If you need to use them, please contact the corresponding author. The original PlanetScope data were obtained through the Planet Education and Research Program, which can be accessed at https://www.planet.com/ (Planet Team, 2019)."* **(P41L713~718)**

**Comment 12**

Lines 274-275 and several other instances contain unclear grammar. It would be beneficial to revise these with the assistance of a native speaker.

**Response 12**

We thank the reviewer for raising these points. We have modified this line.

"*In the aforementioned nine events, the available public data primarily focuses on geological analysis rather than tasks related to semantic segmentation.*" **(P13L292~293)**

In addition, we also corrected other grammatical errors in the manuscript.

**Comment 13**

Additionally, validating the results with data from the recent Taiwan earthquake is suggested.

**Response 13**

Thank you for the suggestion. In the revised manuscript, we have added a new Section 6.4, which details the application of the GDCLD-SegFormer model to the two recent events in 2024: rainfall-induced landslides in Meizhou, and earthquake-induced landslides in Hualien.

"*6.4 Practical Applications of GDCLD*

*To evaluate the practical applicability of the CDCLD, we selected two significant landslide events that occurred in April 2024 for rapid identification. These events include landslides induced by a heavy rainfall in Meizhou, China and landslides triggered by an earthquake in Hualien, China. PlanetScope image was employed in both cases for*

*experimentation. For the Meizhou case, we obtained the image on May 14, 2024, and applied SegFormer model trained on GDCLD to identify landslides triggered by the heavy rainfall. The results, shown in Figure.14, demonstrate that the GDCLD-trained model effectively mapped newly-induced landslides with a total area of 8.49 $km^2$. The model exhibited excellent accuracy in avoiding false positives such as buildings, roads, and rivers. In terms of the Hualien event, we acquired post-event images from April 17 to 29, 2024. The rapid identification results, displayed in Figure.15, indicate that. the GDCLD-trained model effectively eliminates false positives, such as roads, buildings, bare land, and rivers, with an identified landslide area of 90.9 $km^2$. The original PlanetScope images and landslide recognitions of the two events are available at https://doi.org/10.5281/zenodo.13612636 (Fang et al., 2024) (P36~39L651~671)*

[Figure]

*Figure.14 Detection results of rainfall-induced landslides for Meizhou, China. (a) is the aerial view of the whole area; (b), (c) and, (d) are the partial details.*

[Figure]

*Figure.15 Detection results of earthquake-triggered landslides for Hualien, China. (a) is the aerial view of the whole area; (b), (c,) and (d) are the partial details.*"

**Reference**

Bhuyan, K., Rana, K., Ferrer, J. V., Cotton, F., Ozturk, U., Catani, F., and Malik, N.: Landslide topology uncovers failure movements, Nature Communications, 15, 2633, 2024.

Marc, O. and Hovius, N.: Amalgamation in landslide maps: effects and automatic detection, Natural Hazards and Earth System Science, 15, 723-733, 2015

Hungr, O., Leroueil, S., and Picarelli, L.: The Varnes classification of landslide types, an update, Landslides, 11, 167-194, 2014.

Xu, Y., Ouyang, C., Xu, Q., Wang, D., Zhao, B., and Luo, Y.: CAS Landslide Dataset: A Large-Scale and Multisensor Dataset for Deep Learning-Based Landslide Detection, Sci Data, 11, 12, 10.1038/s41597-023-02847-z, 2024.